# EXTENDING THE WILDS BENCHMARK FOR UNSUPERVISED ADAPTATION

**Shiori Sagawa**[1*]**, Pang Wei Koh**[1*]**, Tony Lee**[1*]**, Irena Gao**[1*]**,**
**Sang Michael Xie**[1]**, Kendrick Shen**[1]**, Ananya Kumar**[1]**, Weihua Hu**[1]**, Michihiro Yasunaga**[1]**,**
**Henrik Marklund**[1]**, Sara Beery**[2]**, Etienne David**[3]**, Ian Stavness**[4]**, Wei Guo**[5]**, Jure Leskovec**[1]**,**
**Kate Saenko**[6]**, Tatsunori Hashimoto**[1]**, Sergey Levine**[7]**, Chelsea Finn**[1]**, Percy Liang**[1]
[*]Equal contribution [1]Stanford University [2]Caltech [3]INRAE [4]University of Saskatchewan
[5]University of Tokyo [6]Boston University [7]University of California, Berkeley

## ABSTRACT

Machine learning systems deployed in the wild are often trained on a source distribution but deployed on a different target distribution. Unlabeled data can be a powerful point of leverage for mitigating these distribution shifts, as it is frequently much more available than labeled data and can often be obtained from distributions beyond the source distribution as well. However, existing distribution shift benchmarks with unlabeled data do not reflect the breadth of scenarios that arise in real-world applications. In this work, we present the WILDS 2.0 update, which extends 8 of the 10 datasets in the WILDS benchmark of distribution shifts to include curated unlabeled data that would be realistically obtainable in deployment. These datasets span a wide range of applications (from histology to wildlife conservation), tasks (classification, regression, and detection), and modalities (photos, satellite images, microscope slides, text, molecular graphs). The update maintains consistency with the original WILDS benchmark by using identical labeled training, validation, and test sets, as well as identical evaluation metrics. We systematically benchmark state-of-the-art methods that use unlabeled data, including domain-invariant, self-training, and self-supervised methods, and show that their success on WILDS is limited. To facilitate method development, we provide an open-source package that automates data loading and contains the model architectures and methods used in this paper. Code and leaderboards are available at `https://wilds.stanford.edu`.

## 1 INTRODUCTION

Distribution shifts—when models are trained on a source distribution but deployed on a different target distribution—are frequent problems for machine learning systems in the wild (Quiñonero-Candela et al., 2009; Geirhos et al., 2020; Koh et al., 2021). In this paper, we focus on the use of unlabeled data to mitigate these shifts. Unlabeled data is a powerful point of leverage as it is more readily available than labeled data and can often be obtained from distributions beyond the source distribution. For example, in the crop detection task in Figure 1, we wish to learn a model that can extrapolate to a set of target domains (farms) (David et al., 2020), and while we only have labeled training examples from some source domains, we have many more unlabeled examples from the source domains, from extra domains, and even directly from the target domains.

Many methods for leveraging unlabeled data have been highly successful on some types of distribution shifts (Berthelot et al., 2021; Zhang et al., 2021). However, the datasets typically used for evaluating these methods do not reflect many of the realistic shifts that might occur in the wild. These evaluations tend instead to focus on shifts between photos and stylized versions like sketches (Li et al., 2017; Venkateswara et al., 2017; Peng et al., 2019) or synthetic renderings (Peng et al., 2018), or between variants of digits datasets like MNIST (LeCun et al., 1998) and SVHN (Netzer et al., 2011). Unfortunately, prior work has shown that methods that work well on one type of shift need not generalize to others (Taori et al., 2020; Djolonga et al., 2020; Xie et al., 2021a; Miller et al., 2021), which raises the question of how well they would work on a wider array of realistic shifts.

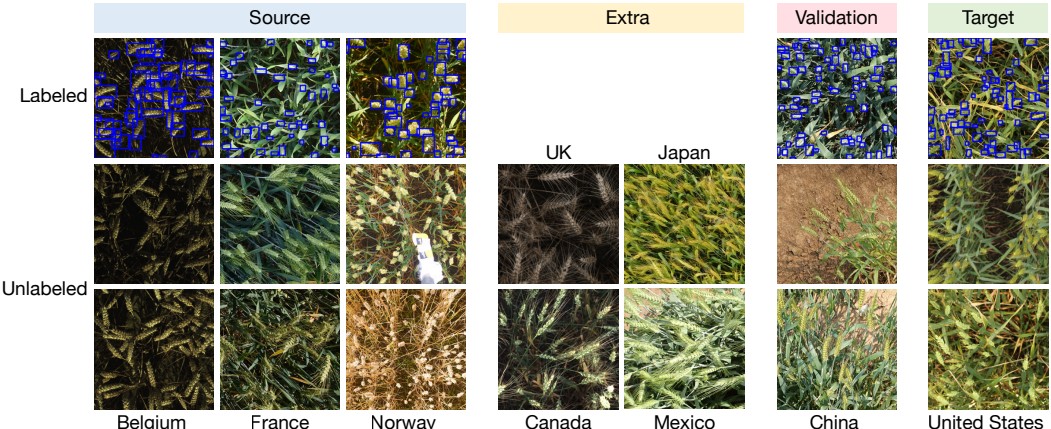

Figure 1: Each WILDS dataset (Koh et al., 2021) contains labeled data from the source domains (for training), validation domains (for hyperparameter selection), and target domains (for held-out evaluation). In the WILDS 2.0 update, we extend these datasets with unlabeled data from a combination of source, validation, or target domains, as well as extra domains from which there is no labeled data. The labeled data is exactly the same as in WILDS 1.0. In this figure, we illustrate the setting with the GLOBALWHEAT-WILDS dataset, where domains correspond to images acquired from different locations and at different times.

In this paper, we make two contributions. First, we present WILDS 2.0 (Figure 2), an updated version of the recent WILDS benchmark of in-the-wild distribution shifts (Koh et al., 2021). WILDS datasets span a wide range of tasks and modalities, and each dataset reflects a domain generalization or subpopulation shift setting with a substantial gap between in-distribution and out-of-distribution performance. However, WILDS 1.0 only contained labeled data, which limits the leverage for learning robust models. In WILDS 2.0, we extend 8 of the 10 WILDS datasets[1] with curated unlabeled data acquired from the same source and target domains as the labeled data, as well as from extra domains of the same type: e.g., in the GLOBALWHEAT-WILDS dataset pictured in Figure 1, we acquired unlabeled photos of wheat fields from the source and target farms as well as extra farms that were not in the original labeled dataset. In total, WILDS 2.0 adds 14.5 million unlabeled examples, expanding the number of examples for each dataset by 3–13× and **allowing us to combine the real-world relevance of WILDS with the leverage of unlabeled data**.

Second, we developed a standardized and consistent protocol for evaluating methods that leverage the unlabeled data in WILDS 2.0. We assessed representatives from three popular categories: methods for learning domain-invariant representations (Sun & Saenko, 2016; Ganin et al., 2016), self-training methods (Lee, 2013; Sohn et al., 2020; Xie et al., 2020), and pre-training methods that rely on self-supervision (Devlin et al., 2019; Caron et al., 2020). These methods have been successful on some types of shifts, such as going from photos to sketches, or from handwritten digits to street signs (Berthelot et al., 2021; Zhang et al., 2021).

**Our results on WILDS are mixed: many methods did not outperform standard supervised training despite using additional unlabeled data**, and the only clear successes were on two image classification datasets (CAMELYON17-WILDS and FMoW-WILDS). Successful methods relied heavily on data augmentation (Xie et al., 2020; Caron et al., 2020), which limited their applicability to modalities where augmentations are not as well developed, such as text and molecular graphs. The same methods were unsuccessful on image regression and detection tasks, which have been relatively understudied: e.g., pseudolabel-based methods do not straightforwardly apply to regression. For the text datasets, continued language model pre-training did not help, unlike in prior work (Gururangan et al., 2020). Our results suggest fruitful avenues for future work, such as developing data augmentations for non-image modalities and more effective hyperparameter tuning protocols.

Overall, our results underscore the importance of developing and evaluating methods for unlabeled data on a wider variety of real-world shifts than is typically studied. To this end, we have updated the open-source Python WILDS package to include unlabeled data loaders, compatible implementations of all the methods we benchmarked, and scripts to replicate all experiments in this paper (Ap-

---

[1] We omitted PY150-WILDS, as code completion data is always labeled by nature of the task, and RXRX1-WILDS, as unlabeled data for that genetic perturbation task is not typically available.

| Dataset | iWildCam | Camelyon17 | RxRx1 | FMoW | PovertyMap | GlobalWheat | OGB-MolPCBA | CivilComments | Amazon | Py150 |
|---|---|---|---|---|---|---|---|---|---|---|
| Input (x) | camera trap photo | tissue slide | cell image | satellite image | satellite image | wheat image | molecular graph | online comment | product review | code |
| Prediction (y) | animal species | tumor | perturbed gene | land use | asset wealth | wheat head bbox | bioassays | toxicity | sentiment | autocomplete |
| Domain (d) | camera | hospital | batch | time, region | country, ru/ur | location, time | scaffold | demographic | user | git repo |
| Source example | | | | | | | | What do Black and LGBT people have to do with bicycle licensing? | Overall a solid package that has a good quality of construction for the price. | import numpy as np … norm=np.___ |
| Target example | | | | | | | | As a Christian, I will not be patronizing any of those businesses. | I "loved" my French press, it's so perfect and came with all this fun stuff! | import subprocess as sp p=sp.Popen() stdout=p.___ |
| Original paper | Beery et al. 2020 | Bandi et al. 2018 | Taylor et al. 2019 | Christie et al. 2018 | Yeh et al. 2020 | David et al. 2021 | Hu et al. 2020 | Borkan et al. 2019 | Ni et al. 2019 | Raychev et al. 2016 |
| **Labeled** # domains | 323 | 5 | 51 | 16 x 5 | 23 x 2 | 47 | 120,084 | 16 | 3,920 | 8,421 |
| # examples | 203,029 | 455,954 | 125,510 | 141,696 | 19,669 | 6,515 | 437,929 | 448,000 | 539,502 | 150,000 |
| **Unlabeled** Source domains # domains | - | 3 | - | 11 x 5 | 13 x 2 | 18 | 44,930 | - | - | - |
| # examples | - | 1,799,247 | - | 11,948 | 181,948 | 5,997 | 4,052,627 | - | - | - |
| Extra domains # domains | 3,215 | - | - | - | - | 53 | - | 1 | 21,694 | - |
| # examples | 819,120 | - | - | - | - | 42,445 | - | 1,551,515 | 2,927,841 | - |
| Validation domains # domains | - | 1 | - | 3 x 5 | 5 x 2 | 11 | 31,361 | - | 1,334 | - |
| # examples | - | 600,030 | - | 155,313 | 24,173 | 2,000 | 430,325 | - | 266,066 | - |
| Target domains # domains | - | 1 | - | 2 x 5 | 5 x 2 | 18 | 43,793 | - | 1,334 | - |
| # examples | - | 600,030 | - | 173,208 | 55,275 | 8,997 | 517,048 | - | 268,761 | - |

Figure 2: The WILDS 2.0 update adds unlabeled data to 8 WILDS datasets. For each dataset, we kept the labeled data from WILDS and expanded the datasets by 3–13× with unlabeled data from the same underlying dataset. The type of unlabeled data (i.e., whether it comes from source, extra, validation, or target domains) depends on what is realistic and available for the application. Beyond these 8 datasets, WILDS also contains 2 datasets without unlabeled data: the PY150-WILDS code completion dataset and the RXRX1-WILDS genetic perturbation dataset. For all datasets, the labeled data and evaluation metrics are exactly the same as in WILDS 1.0. Figure adapted with permission from Koh et al. (2021).

pendix G). Code and public leaderboards are available at https://wilds.stanford.edu. By allowing developers to easily test algorithms across the variety of datasets in WILDS 2.0, we hope to accelerate the development of methods that can leverage unlabeled data to improve robustness to real-world distribution shifts.

Finally, we note that WILDS 2.0 not a separate benchmark from WILDS 1.0: the labeled data and evaluation metrics are exactly the same in WILDS 1.0 and WILDS 2.0, and future results should be reported on the overall WILDS benchmark, with a note describing what kind of unlabeled data (if any) was used. In this paper, we discuss the addition of unlabeled data and analyze the performance of methods that use the unlabeled data. For a more detailed description of the datasets, evaluation metrics, and models used, please refer to the original WILDS paper (Koh et al., 2021).

## 2 COMPARISON WITH EXISTING UNSUPERVISED ADAPTATION BENCHMARKS

WILDS 2.0 offers a diverse range of applications and modalities while also providing an extensive amount of unlabeled data that can be used as leverage for training robust models. In this section, we briefly compare with other existing ML benchmarks for unsupervised adaptation.

**Images.** Evaluations of unsupervised adaptation methods for image classification have focused on generalizing from natural photos to a range of stylized images, such as sketches and cartoons (PACS (Li et al., 2017), Office-Home (Venkateswara et al., 2017), and DomainNet (Peng et al., 2019)), product images (Office-31 (Saenko et al., 2010)), and synthetic renderings (VisDA (Peng et al., 2018)), though location-based shifts have also been recently explored (Dubey et al., 2021). It is also popular to evaluate on shifts between digits datasets, such as MNIST (LeCun et al., 1998), SVHN (Netzer et al., 2011), and USPS (Hull, 1994). In image detection and segmentation, existing adaptation benchmarks tend to focus on generalizing from synthetic to natural scenes (Ros et al., 2016; Richter et al., 2016; Cordts et al., 2016; Hoffman et al., 2018), which can be an important tool for realistic problems but is not the focus of this work. In contrast, WILDS considers real-world distribution shifts, and it spans diverse modalities (satellite, microscope, agriculture, and camera trap images) and tasks (classification, regression, detection).

**Text.** Methods for unsupervised adaptation in NLP are typically evaluated on domain shifts between different textual sources, such as news articles, different categories of product reviews, Wikipedia,

or social media platforms (Blitzer et al., 2007; Mansour et al., 2009; Oren et al., 2019; Miller et al., 2020; Kamath et al., 2020; Hendrycks et al., 2020), or even more specialized sources such as legal documents (Chalkidis et al., 2020) or biomedical papers (Lee et al., 2020b; Gu et al., 2020). Multilingual tasks can also be a setting for unsupervised adaptation (Conneau et al., 2018; Conneau & Lample, 2019; Hu et al., 2020a; Clark et al., 2020), especially when generalizing to low-resource languages (Nekoto et al., 2020). The WILDS text datasets differ in that they focus on subpopulation performance, either to particular demographics in CIVILCOMMENTS-WILDS or to tail populations in AMAZON-WILDS, rather than on adapting to a completely distinct domain.

**Molecules.** While unlabeled molecules have been used for pre-training (Hu et al., 2020c; Rong et al., 2020), no standardized unsupervised adaptation benchmarks have been developed.

## 3 PROBLEM SETTING

As in WILDS 1.0, we study the domain shift setting where the data is drawn from domains $d \in \mathcal{D}$. Each domain $d$ corresponds to a data distribution $P_d$ over $(x, y, d)$, where $x$ is the input, $y$ is the prediction, and all points from $P_d$ have domain $d$. See Koh et al. (2021) for more details. The domains come in four types:

| Type of domain | Labeled data | Unlabeled data |
|---|---|---|
| Source domains | Used for training | |
| Extra domains | None | Can be used for training, if available |
| Validation domains | Used for hyperparameter tuning | |
| Target domains | Used for held-out evaluation | |

Table 1: All datasets have labeled source, validation, and target data, as well as unlabeled data from one or more types of domains, depending on what is realistic for the application.

We consider several variants of the domain shift setting. In some applications, all four types of domains are disjoint (e.g., if we are training on labeled data from some hospitals but seeking to generalize to new hospitals); in others, the target domains are a subset of the source domains (e.g., if we are training on a heterogeneous dataset but seeking to measure model performance on particular demographic subpopulations). Models are trained on labeled data from the source domains, as well as unlabeled data of one or more types of domains, depending on what is realistic for the application.

## 4 DATASETS

WILDS 2.0 augments 8 WILDS datasets with curated unlabeled data. For consistency, the labeled datasets and evaluation metrics are exactly the same as in WILDS 1.0, which allows direct evaluations of the utility of unlabeled training data. The labeled and unlabeled data are disjoint, e.g., the unlabeled target data is different from the labeled target data used for evaluation. Here, we briefly describe each dataset, why unlabeled data can be realistically obtained for the corresponding task, and how it might help. In Appendix A, we provide more information on each dataset, including data provenance and details on data processing. In general, all of the unlabeled datasets in WILDS 2.0 were processed in a similar way as their corresponding labeled datasets from WILDS 1.0.

**IWILDCAM2020-WILDS: Species classification across different camera traps.** The task is to classify the animal species in a camera trap image (Beery et al., 2020). We aim to generalize to new camera trap locations despite variations in illumination, background, and label frequencies (Beery et al., 2018). While hundreds of thousands of camera traps are active worldwide, only a small subset of these traps have had images labeled, and the unlabeled data from the other camera traps capture diverse operating conditions that can be used to learn robust models. In this work, we add unlabeled images from 3,215 extra camera traps also in the WCS Camera Traps dataset (Beery et al., 2020). This expands the number of camera traps by $11\times$ and the number of examples by $5\times$.

**CAMELYON17-WILDS: Tumor identification across different hospitals.** The task is to classify image patches from lymph node sections as tumor or normal tissue. We seek to generalize to new hospitals, which can differ in their patient demographics and data acquisition protocols (Veta

et al., 2016; AlBadawy et al., 2018; Komura & Ishikawa, 2018; Tellez et al., 2019). While obtaining labeled data for histopathology applications requires pain-staking annotations from expert pathologists, hospitals typically accumulate unlabeled slide images during normal operation. These unlabeled images could be used to adapt to differences between hospitals (e.g., different staining protocols might lead to different color distributions). We provide unlabeled patches from train and test hospitals, which expands the total number of patches by $7.5\times$. Both the labeled and unlabeled data are adapted from the Camelyon17 dataset (Bandi et al., 2018).

**FMoW-WILDS: Land use classification across different regions and years.** The task is to classify the type of building or land usage in a satellite image. Given training data from before 2013, we aim to generalize to satellite imagery taken after 2013, while maintaining high accuracy across all geographic regions. While labeling land use requires combining map data and expert annotations, unlabeled data is available in all locations in the world through constant streams of global satellite imagery. Prior work has shown that unlabeled satellite data can improve OOD accuracy in landcover and cropland prediction (Xie et al., 2021a) as well as aerial object and scene classification (Reed et al., 2021). We provide unlabeled satellite imagery across all regions from the train and test timeframes defined in WILDS, expanding the dataset by $3.5\times$. Both the labeled and unlabeled data are adapted from the FMoW dataset (Christie et al., 2018).

**POVERTYMAP-WILDS: Poverty mapping across different countries.** The task is to predict a real-valued asset wealth index of the area in a satellite image. We consider generalizing across different countries. Like FMoW-WILDS, unlabeled satellite imagery is available globally, while labeled data is expensive to collect as it requires conducting nationally representative surveys in the field. Prior work on poverty prediction has used unlabeled data for entropy minimization (Jean et al., 2018) and pre-training on auxiliary tasks such as nighttime light prediction (Xie et al., 2016; Jean et al., 2016), but these studies do not study generalization to new countries. We provide unlabeled satellite imagery from both train and test countries, expanding the dataset by $14\times$. Both the labeled and unlabeled data are adapted from Yeh et al. (2020).

**GLOBALWHEAT-WILDS: Wheat head detection across different regions.** The task is to localize wheat heads in overhead field images. We seek to generalize across image acquisition sessions, each of which represents a particular location, time, and sensor; these can differ in wheat genotype, wheat head appearance, growing conditions, background appearance, illumination, and acquisition protocols. Wheat field images contain many densely packed and overlapping instances, making labeling wheat heads in images costly, tedious and sensitive to the individual annotator. However, hundreds of agricultural research institutes around the world collect terabytes of unlabeled field images which could be used for training. We add unlabeled field images from train, test, and extra acquisition sessions, expanding the dataset by $10\times$. The labeled and unlabeled data are adapted from the Global Wheat Head Detection dataset and its underlying sources (David et al., 2020; 2021).

**OGB-MOLPCBA: Molecular property prediction across different scaffolds.** The task is to predict the biological activity of small molecules represented as molecular graphs (Wu et al., 2018; Hu et al., 2020b). We seek to generalize to molecules with new scaffold structures. Labels on biological activity are only available for a small portion of molecules, as they require expensive lab experiments to obtain. However, unlabeled molecule structures are readily available in large-scale chemical databases such as PubChem (Bolton et al., 2008), and have been previously used for pre-training (Hu et al., 2020c) and semi-supervised learning (Sun et al., 2020). We provide 5 million unlabeled molecules from source and target scaffolds, which expands the number of molecules by $12.5\times$. The original labeled data was curated by MoleculeNet (Wu et al., 2018) from PubChem, and we similarly extracted the unlabeled data from PubChem (Bolton et al., 2008).

**CIVILCOMMENTS-WILDS: Toxicity classification across demographic identities.** The task is to classify whether a text comment is toxic or not. We consider the subpopulation shift setting, where the model must classify accurately across groups of comments mentioning different demographic identities. While labels require large-scale crowdsourcing annotations on both comment toxicity, unlabeled article comments are widely available on the internet. We provide unannotated comments as unlabeled data, which expands the size of the dataset by $4.5\times$. Both the labeled and unlabeled data are adapted from Borkan et al. (2019).

**AMAZON-WILDS: Sentiment classification across different users.** The task is to classify the star ratings of Amazon reviews. We seek to perform consistently well across new reviewers. While the

labels (star ratings) are always available for Amazon reviews in practice, unlabeled data is a common source of leverage for sentiment classification more generally, with prior work in domain adaptation (Blitzer & Pereira, 2007; Glorot et al., 2011) and semi-supervised learning (Dasgupta & Ng, 2009; Li et al., 2011). We provide unlabeled reviews from test and extra reviewers, which expands the total number of reviews by $7.5\times$. Both the labeled and unlabeled data are adapted from the Amazon review dataset by Ni et al. (2019).

## 5 ALGORITHMS

For our evaluation, we selected representative methods from the three categories described below. These methods exemplify current approaches to using unlabeled data to improve robustness, and they have been successful on popular domain adaptation benchmarks like DomainNet (Peng et al., 2019) and semi-supervised settings like improving ImageNet accuracy by leveraging unlabeled images from the internet (Xie et al., 2020; Caron et al., 2020). For more details, see Appendix B.

**Domain-invariant methods.** Domain-invariant methods learn feature representations that are invariant across different domains by penalizing differences between learned source and target representations (Long et al., 2015; Ganin et al., 2016; Sun & Saenko, 2016; Long et al., 2017; 2018; Saito et al., 2018; Zhang et al., 2018; Xu et al., 2019; Zhang et al., 2019b). We discuss these methods further in Appendix B.2. For our experiments, we evaluate two classical methods:

- *Domain-Adversarial Neural Networks (DANN)* (Ganin et al., 2016) penalize representations on which an auxiliary classifier can easily discriminate between source and target examples.

- *Correlation Alignment (CORAL)* (Sun et al., 2016; Sun & Saenko, 2016) penalizes differences between the means and covariances of the source and target feature distributions.

**Self-training.** Self-training methods "pseudo-label" unlabeled examples with the model's own predictions and then train on them as if they were labeled examples. These methods often also use consistency regularization, which encourages the model to make consistent predictions on augmented views of unlabeled examples (Sohn et al., 2020; Xie et al., 2020; Berthelot et al., 2021). Self-training methods have recently been successfully applied to unsupervised adaptation (Saito et al., 2017; Berthelot et al., 2021; Zhang et al., 2021). We include three representative algorithms:

- *Pseudo-Label* (Lee, 2013) dynamically generates pseudolabels and updates the model each batch.

- *FixMatch* (Sohn et al., 2020) adds consistency regularization on top of the Pseudo-Label algorithm. Specifically, it generates pseudolabels on a weakly augmented view of the unlabeled data, and then minimizes the loss of the model's prediction on a strongly augmented view.

- *Noisy Student* (Xie et al., 2020) leverages weak and strong augmentations like FixMatch, but instead of dynamically generating pseudolabels for each batch, it alternates between a few teacher phases, where it generates pseudolabels, and student phases, where it trains to convergence on the (pseudo)labeled data.

**Self-supervision.** Self-supervised methods learn useful representations by training on unlabeled data via auxiliary proxy tasks. Common approaches include reconstruction tasks (Vincent et al., 2008; Erhan et al., 2010; Devlin et al., 2019; Gidaris et al., 2018; Lewis et al., 2020), and contrastive learning (He et al., 2020; Chen et al., 2020b; Caron et al., 2020; Radford et al., 2021b), and recent work has shown that self-supervised methods can reduce dependence on spurious correlations and improve performance on domain adaptation tasks (Wang et al., 2021; Tsai et al., 2021; Mishra et al., 2021). We use these self-supervision methods for unsupervised adaptation by first pre-training models on the unlabeled data, and then finetuning them on the labeled source data (Shen et al., 2021). We evaluate popular self-supervised methods for vision and language:

- *SwAV* (Caron et al., 2020) is a contrastive learning algorithm that maps representations to a set of clusters and then enforces similarity between cluster assignments.

- *Masked language modeling (MLM)* (Devlin et al., 2019) randomly masks some of the tokens from input text and trains the model to predict the missing tokens.

## 6 EXPERIMENTS

To evaluate how well existing methods can leverage unlabeled data to be robust to in-the-wild distribution shifts, we benchmarked the methods above on all applicable WILDS 2.0 datasets.

### 6.1 SETUP

We used the default models, labeled training and test sets, and evaluation metrics from WILDS.

**Unlabeled data.** WILDS 2.0 contains multiple types of unlabeled data (from source, extra, validation, and/or target domains). For simplicity, we ran experiments on a single type of unlabeled data for each dataset. Where possible, we used unlabeled target data to allow methods to directly adapt to the target distribution; for IWILDCAM2020-WILDS and CIVILCOMMENTS-WILDS, which do not have unlabeled target data, we used the extra domains instead. All methods use exactly the same sets of labeled and unlabeled training data (except ERM, which does not use unlabeled data).

**Hyperparameters.** We tuned each method on each dataset separately using random hyperparameter search. Following WILDS 1.0, we used the labeled out-of-distribution (OOD) validation set to select hyperparameters and for early stopping (Koh et al., 2021). This validation set is drawn from a different distribution than both the training and the OOD test set, so tuning on it does not leak information on the test distribution. We did not use the in-distribution (ID) validation set. For image classification and regression, we used both RandAugment (Cubuk et al., 2020) and Cutout (DeVries & Taylor, 2017) as data augmentation for all methods. We did not use data augmentation for the remaining datasets. For some datasets, we also had ground truth labels for the "unlabeled" data, which we used to run fully-labeled ERM experiments. Overall, we ran 600+ experiments for 7,000 GPU hours on NVIDIA V100s. See Appendix B for a discussion of which methods were applicable to which datasets; Appendix C for augmentation details; Appendix F for the fully-labeled experiments; Appendix D for further experimental details.

### 6.2 RESULTS

Table 2 shows mixed results on WILDS: most methods do not improve over standard empirical risk minimization (ERM) despite access to unlabeled data and careful hyperparameter tuning. In contrast, these methods have been shown to perform well on prior unsupervised adaptation benchmarks; in Appendix E, we verify our implementations by showing that these methods (with the exception of CORAL) outperform ERM on the *real → sketch* shift in DomainNet, a standard unsupervised adaptation benchmark for object classification (Peng et al., 2019).

**Image classification (IWILDCAM2020-WILDS, CAMELYON17-WILDS, and FMOW-WILDS).** Data augmentation improved OOD performance on all three image classification datasets. The gain was the most substantial on CAMELYON17-WILDS, where vanilla ERM achieved 70.8% accuracy, while ERM with data augmentation achieved 82.0% accuracy.[2]

On CAMELYON17-WILDS and FMOW-WILDS, where we had access to unlabeled target data, Noisy Student and SwAV pre-training consistently improved OOD performance and reduced variability across replicates. However, the other methods—CORAL, DANN, Pseudo-Label, and FixMatch—underperformed ERM. This was especially surprising for FixMatch, which performed very well on DomainNet (Appendix E). Both FixMatch and Noisy Student use pseudo-labeling and consistency regularization, but FixMatch dynamically computes pseudo-labels in each batch from the start of training, whereas Noisy Student first trains a teacher model to convergence on the labeled data and updates pseudolabels at a much slower rate. As in Xie et al. (2020), this suggests that dynamically updating pseudo-labels might hurt generalization.

On IWILDCAM2020-WILDS, where we had access to $4\times$ as many unlabeled images from extra domains (distinct camera traps) but not to any images from the target domains, none of the benchmarked methods improved OOD performance compared to ERM. This was surprising, as many of these methods were originally shown to work in semi-supervised settings. One difference could be that the labeled and unlabeled examples in IWILDCAM2020-WILDS differ more significantly (as

---

[2]The data augmentation involves color jitter, which simulates the difference in staining protocols between the source and target distributions in CAMELYON17-WILDS (Koh et al., 2021; Robey et al., 2021).

| | iWildCam2020-wilds (Unlabeled extra, macro F1) | | FMoW-wilds (Unlabeled target, worst-region acc) | |
|---|---|---|---|---|
| | In-distribution | Out-of-distribution | In-distribution | Out-of-distribution |
| ERM (-data aug) | 46.7 (0.6) | 30.6 (1.1) | 59.3 (0.7) | 33.7 (1.5) |
| ERM | 47.0 (1.4) | **32.2** (1.2) | 60.6 (0.6) | 34.8 (1.5) |
| CORAL | 40.5 (1.4) | 27.9 (0.4) | 58.9 (0.3) | 34.1 (0.6) |
| DANN | 48.5 (2.8) | **31.9** (1.4) | 57.9 (0.8) | 34.6 (1.7) |
| Pseudo-Label | 47.3 (0.4) | 30.3 (0.4) | 60.9 (0.5) | 33.7 (0.2) |
| FixMatch | 46.3 (0.5) | **31.0** (1.3) | 58.6 (2.4) | 32.1 (2.0) |
| Noisy Student | 47.5 (0.9) | **32.1** (0.7) | 61.3 (0.4) | **37.8** (0.6) |
| SwAV | 47.3 (1.4) | 29.0 (2.0) | 61.8 (1.0) | 36.3 (1.0) |
| ERM (fully-labeled) | 54.6 (1.5) | 44.0 (2.3) | 65.4 (0.4) | 58.7 (1.4) |

| | Camelyon17-wilds (Unlabeled target, avg acc) | | PovertyMap-wilds (Unlabeled target, worst U/R corr) | |
|---|---|---|---|---|
| | In-distribution | Out-of-distribution | In-distribution | Out-of-distribution |
| ERM (-data aug) | 85.8 (1.9) | 70.8 (7.2) | 0.65 (0.03) | **0.50** (0.07) |
| ERM | 90.6 (1.2) | 82.0 (7.4) | 0.66 (0.04) | **0.49** (0.06) |
| CORAL | 90.4 (0.9) | 77.9 (6.6) | 0.54 (0.10) | 0.36 (0.08) |
| DANN | 86.9 (2.2) | 68.4 (9.2) | 0.50 (0.07) | 0.33 (0.10) |
| Pseudo-Label | 91.3 (1.3) | 67.7 (8.2) | – | – |
| FixMatch | 91.3 (1.1) | 71.0 (4.9) | 0.54 (0.11) | 0.30 (0.11) |
| Noisy Student | 93.2 (0.5) | 86.7 (1.7) | 0.61 (0.07) | 0.42 (0.11) |
| SwAV | 92.3 (0.4) | **91.4** (2.0) | 0.60 (0.13) | **0.45** (0.05) |

| | GlobalWheat-wilds (Unlabeled target, avg domain acc) | | OGB-MolPCBA (Unlabeled target, avg AP) | |
|---|---|---|---|---|
| | In-distribution | Out-of-distribution | In-distribution | Out-of-distribution |
| ERM | 77.8 (0.1) | **50.5** (1.7) | – | **28.3** (0.1) |
| CORAL | – | – | – | 26.6 (0.2) |
| DANN | – | – | – | 20.4 (0.8) |
| Pseudo-Label | 75.2 (1.2) | 42.7 (4.8) | – | 19.7 (0.1) |
| Noisy Student | 78.8 (0.5) | **49.3** (3.7) | – | 27.5 (0.1) |

| | CivilComments-wilds (Unlabeled extra, worst-group acc) | | Amazon-wilds (Unlabeled target, 10th percentile acc) | |
|---|---|---|---|---|
| | In-distribution | Out-of-distribution | In-distribution | Out-of-distribution |
| ERM | 89.8 (0.8) | **66.6** (1.6) | 72.0 (0.1) | **54.2** (0.8) |
| CORAL | – | – | 71.7 (0.1) | 53.3 (0.0) |
| DANN | – | – | 71.7 (0.1) | 53.3 (0.0) |
| Pseudo-Label | 90.3 (0.5) | **66.9** (2.6) | 71.6 (0.1) | 52.3 (1.1) |
| Masked LM | 89.4 (1.2) | **65.7** (2.3) | 71.9 (0.4) | **53.9** (0.7) |
| ERM (fully-labeled) | 89.9 (0.1) | 69.4 (0.6) | 73.6 (0.1) | 56.4 (0.8) |

Table 2: The in-distribution (ID) and out-of-distribution (OOD) performance of each method on each applicable dataset. Following WILDS 1.0, we ran 3–10 replicates (random seeds) for each cell, depending on the dataset. We report the standard deviation across replicates in parentheses; the standard error (of the mean) is lower by the square root of the number of replicates. Fully-labeled experiments use ground truth labels on the "unlabeled" data. We bold the highest non-fully-labeled OOD performance numbers as well as others where the standard error is within range. Below each dataset name, we report the type of unlabeled data and metric used.

they originate from different camera traps) than in the original FixMatch paper (Sohn et al., 2020), which used i.i.d. labeled and unlabeled data, or the Noisy Student paper (Xie et al., 2020), which used ImageNet labeled data (Russakovsky et al., 2015) and JFT unlabeled data (Hinton et al., 2015).

Fully-labeled ERM models that used ground truth labels for the "unlabeled" data were available for FMoW-wilds and iWildCam2020-wilds. They significantly outperformed other methods, suggesting room for improvement in how we leverage the unlabeled data.

**Image regression (PovertyMap-wilds).** Data augmentation had no effect on performance on PovertyMap-wilds, which differs from the above image datasets in that it is a regression task and involves multi-spectral satellite images (with 7 channels); both of these aspects are relatively unstudied compared to standard RGB image classification. All applicable methods underperformed standard ERM, despite having access to unlabeled data from the target domains (countries).

**Image detection (GLOBALWHEAT-WILDS).** We did not apply data augmentation here, as standard augmentation changes the labels (e.g., cropping the image might remove bounding boxes) and would violate the assumption that labels are invariant under augmentations, which contrastive and consistency regularization methods like SwAV, Noisy Student, and FixMatch rely on. Accordingly, we did not evaluate FixMatch and SwAV, and we modified Noisy Student to remove data augmentation noise. All applicable methods underperformed ERM.

**Molecule classification (OGB-MOLPCBA).** We did not apply data augmentation techniques to OGB-MOLPCBA as they are not well-developed for molecular graphs. All methods underperformed ERM. We did not report ID results as this dataset has no separate ID test set.

**Text classification (CIVILCOMMENTS-WILDS, AMAZON-WILDS).** Likewise, we did not apply data augmentation to the text datasets. On both datasets, other methods performed similarly to ERM (with class-balancing for CIVILCOMMENTS-WILDS). Continued masked LM pre-training on the unlabeled data did not improve target performance, unlike in prior work (Gururangan et al., 2020); this might be because the BERT pre-training corpus (Devlin et al., 2019; Hendrycks et al., 2020) is more similar to the online comments in CIVILCOMMENTS-WILDS and product reviews in AMAZON-WILDS than to the biomedical/CS text studied in Gururangan et al. (2020). Also, CIVILCOMMENTS-WILDS and AMAZON-WILDS measure subpopulation performance (on minority demographics and on the tail subpopulation, respectively), whereas prior work adapted models to new areas of the input space (e.g., from news to biomedical articles). Fully-labeled ERM models showed modest gains compared to FMOW-WILDS and iWILDCAM2020-WILDS. As our evaluations on these text datasets focus on subpopulations performance, these results are consistent with prior observations that ERM models can have poor subpopulation performance even with large labeled training sets (Sagawa et al., 2020), necessitating other approaches to subpopulation shifts.

## 7 DISCUSSION

We conclude by discussing several takeaways and promising directions for future work.

**The role of data augmentation.** Many unsupervised adaptation methods rely strongly on data augmentation for consistency regularization or contrastive learning. This reliance on data augmentation techniques—which are largely image-specific—restricts their generality, as they do not readily generalize to other modalities (or even other types of images besides photos). Developing data augmentation techniques that can work well in other applications and modalities could be crucial for expanding the applicability of these methods (Verma et al., 2021).

**Hyperparameter tuning.** Unsupervised adaptation methods have even more hyperparameters than standard supervised methods, and consistent with prior work, we found that these hyperparameters can significantly affect OOD performance (Saito et al., 2021). Moreover, unlike in standard i.i.d. settings, we do not have labeled target data that we can use for hyperparameter selection. Improved methods for hyperparameter tuning could significantly improve OOD performance. Such methods might make use of the unlabeled target data, or even the combination of labeled and unlabeled OOD validation data, which is provided for most datasets in WILDS 2.0.

**Pre-training on broader unlabeled data.** Pre-training on huge amounts of unlabeled data improves robustness to distribution shifts in some settings (Bommasani et al., 2021). The unlabeled data need not be related to the task: e.g., CLIP was pre-trained on text-image pairs from the internet but tested on tasks including histopathology and satellite image classification (Radford et al., 2021a). Existing techniques for this type of broad pre-training appear insufficient for WILDS: many of our models were initialized with ImageNet-pretrained weights or derivatives of BERT, but do not generalize well OOD. While we focused on providing curated unlabeled data that is closely tailored to the task, it could be fruitful to use both broad and curated unlabeled data.

**Leveraging domain annotations and task-specific structure.** OOD robustness is ill-posed in general, as models cannot be robust to arbitrary distribution shifts. Beyond unlabeled data, WILDS also has domain annotations and other structured metadata for both labeled and unlabeled data (e.g., in iWILDCAM2020-WILDS, we know which images were taken from which cameras). Exploiting this type of fine-grained domain structure for unsupervised adaptation—e.g., through multi-source/multi-target domain adaptation methods (Zhao et al., 2018; Peng et al., 2019)—could be a promising avenue for learning models that are more robust to the domain shifts in WILDS.

ETHICS STATEMENT

All WILDS datasets are curated and adapted from public data sources, with licenses that allow for public release. The datasets are all anonymized.

The distribution shifts in several of the WILDS datasets deal with issues of discrimination and bias that arise in real-world applications. For example, CIVILCOMMENTS-WILDS studies disparate model performance across online comments that mention different demographic groups, while FMOW-WILDS and POVERTYMAP-WILDS study countries and regions where labeled satellite data is less readily available. As our results suggest, standard models trained on these datasets will not perform well on those subpopulations, and their learned representations might also be biased in undesirable ways (Bolukbasi et al., 2016; Caliskan et al., 2017; Garg et al., 2018; Tan & Celis, 2019; Steed & Caliskan, 2021). We also encourage caution in interpreting positive results on these datasets, as our evaluation metrics might not encompass all relevant facets of discrimination and bias: e.g., the "ground truth" toxicity annotations in CIVILCOMMENTS-WILDS can themselves be biased, and the particular choice of regions in FMOW-WILDS might obscure lower model performance in sub-regions.

For FMOW-WILDS and POVERTYMAP-WILDS, surveillance and privacy issues also need to be considered. In FMOW-WILDS, the image resolution is lower than that of other public satellite data (e.g., from Google Maps), and in POVERTYMAP-WILDS, the location metadata is noised to protect privacy. For a deeper discussion of the ethics of remote sensing in the context of humanitarian aid and development, we refer readers to the UNICEF report by Berman et al. (2018).

REPRODUCIBILITY STATEMENT

All WILDS datasets are publicly available at https://wilds.stanford.edu, together with code and scripts to replicate all of the experiments in this paper. We also provide all trained model checkpoints and results, together with the exact hyperparameters used.

In our appendices, we provide more details on the datasets and experiments:

- In Appendix A, we describe each of the updated datasets in WILDS 2.0 and their sources of unlabeled data as well as what data processing steps were taken.

- In Appendix B, we describe the implementations of each of our benchmarked methods in detail. In particular, we discuss any changes we made to their original implementations, either for consistency with other methods or with prior implementations of these methods.

- In Appendix C, we describe details of the data augmentations (if any) that we used across each dataset.

- In Appendix D, we describe our experimental protocol, including the hyperparameter selection procedure and hyperparameter grids for all of the methods and datasets.

- In Appendix E, we describe the details of our experiments on DomainNet.

- In Appendix F, we describe the details of our fully-labeled ERM experiments.

- Finally, in Appendix G, we include an illustrative code snippet of how to use the data loaders in the WILDS library.

AUTHOR CONTRIBUTIONS

The project was initiated by Shiori Sagawa, Pang Wei Koh, and Percy Liang. Shiori Sagawa and Pang Wei Koh led the project and coordinated the activities below. Tony Lee developed the experimental infrastructure and ran the experiments. Tony Lee, Irena Gao, Sang Michael Xie, Kendrick Shen, Ananya Kumar, and Michihiro Yasunaga designed the evaluation framework and implemented the algorithms. The unlabeled data loaders and corresponding dataset writeups were added by:

- AMAZON-WILDS: Tony Lee
- CAMELYON17-WILDS: Tony Lee

- CIVILCOMMENTS-WILDS: Irena Gao
- FMoW-WILDS: Sang Michael Xie
- IWILDCAM2020-WILDS: Henrik Marklund and Sara Beery
- OGB-MoLPCBA: Weihua Hu
- POVERTYMAP-WILDS: Sang Michael Xie
- GLOBALWHEAT-WILDS: Etienne David, Ian Stavness, and Wei Guo.

Tony Lee and Henrik Marklund set up the website and leaderboards. Jure Leskovec, Kate Saenko, Tatsunori Hashimoto, Sergey Levine, Chelsea Finn and Percy Liang provided advice on the overall project direction and experimental design and analysis throughout. Shiori Sagawa, Pang Wei Koh, and Irena Gao drafted the paper; all authors contributed towards writing the final paper.

## ACKNOWLEDGEMENTS

We would like to thank Ashwin Ramaswami, Berton Earnshaw, Bowen Liu, Hongseok Namkoong, Junguang Jiang, Ludwig Schmidt, Robbie Jones, Robin Jia, Ruijia Xu, and Yabin Zhang for their helpful advice.

The design of the WILDS benchmark was inspired by the Open Graph Benchmark (Hu et al., 2020b), and we are grateful to the Open Graph Benchmark team for their advice and help in setting up our benchmark.

This project was funded by an Open Philanthropy Project Award and NSF Award Grant No. 1805310. Shiori Sagawa was supported by the Herbert Kunzel Stanford Graduate Fellowship and the Apple Scholars in AI/ML PhD fellowship. Sang Michael Xie was supported by a NDSEG Graduate Fellowship. Ananya Kumar was supported by the Rambus Corporation Stanford Graduate Fellowship. Weihua Hu was supported by the Funai Overseas Scholarship and the Masason Foundation Fellowship. Michihiro Yasunaga was supported by the Microsoft Research PhD Fellowship. Henrik Marklund was supported by the Dr. Tech. Marcus Wallenberg Foundation for Education in International Industrial Entrepreneurship, CIFAR, and Google. Sara Beery was supported by an NSF Graduate Research Fellowship and is a PIMCO Fellow in Data Science. Jure Leskovec is a Chan Zuckerberg Biohub investigator. Chelsea Finn is a CIFAR Fellow in the Learning in Machines and Brains Program.

We also gratefully acknowledge the support of DARPA under Nos. N660011924033 (MCS); ARO under Nos. W911NF-16-1-0342 (MURI), W911NF-16-1-0171 (DURIP); NSF under Nos. OAC-1835598 (CINES), OAC-1934578 (HDR), CCF-1918940 (Expeditions), IIS-2030477 (RAPID); Stanford Data Science Initiative, Wu Tsai Neurosciences Institute, Chan Zuckerberg Biohub, Amazon, JPMorgan Chase, Docomo, Hitachi, JD.com, KDDI, NVIDIA, Dell, Toshiba, and UnitedHealth Group.

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

APPENDICES

# A    ADDITIONAL DATASET DETAILS

In this appendix, we provide additional details on the unlabeled data in WILDS 2.0. For more context on the motivation behind each dataset, the choice of evaluation metric, and the labeled data, please refer to the original WILDS paper (Koh et al., 2021).

## A.1    IWILDCAM2020-WILDS

The IWILDCAM2020-WILDS dataset was adapted from the iWildCam 2020 competition dataset made up of data provided by the Wildlife Conservation Society (WCS) (Beery et al., 2020) [3]. Camera trap images are captured by motion-triggered static cameras placed in the wild to study wildlife in a non-invasive manner. Images are captured at high volumes – a single camera trap can capture 10K images in a month – and annotating these images requires species identification expertise and is time-intensive. However, there are tens of thousands of camera traps worldwide capturing images of wildlife that could be used as unlabeled training data. For example, Wildlife Insights (Ahumada et al., 2020) now contains almost 20M camera trap images collected across the globe, but a large proportion of that data is still unlabeled. Ideally we could capture value from those images despite the lack of available labels. We extend IWILDCAM2020-WILDS with unlabeled data from a set of WCS camera traps entirely disjoint with the labeled dataset, representative of unlabeled data from a newly-deployed sensor network.

**Problem setting.**    The task is to classify the species of animals in camera trap images. The input $x$ is an image from a camera trap, and the domain $d$ corresponds to the camera trap that captured the image. The target $y$, provided only for the labeled training images, is one of 182 classes of animals. We seek to learn models that generalize well to new camera trap deployments, so the test data comes from domains unseen during training. Additionally, we evaluate the in-distribution performance on held-out images from camera traps in the train set.

**Data.**    The data comes from multiple camera traps around the world, all provided by the Wildlife Conservation Society (WCS). The labeled data is the same as in Koh et al. (2021) and the unlabeled data comprise 819,120 images from 3215 WCS camera traps not included in iWildCam 2020:

1. **Source**: 243 camera traps.
2. **Validation (OOD):** 32 camera traps.
3. **Target (OOD):** 48 camera traps.
4. **Extra:** 3215 camera traps.

The four sets of camera traps are disjoint. The distributions of the labeled and unlabeled camera traps are very similar, except that the labeled data does not contain cameras with photos taken before LandSat 8 data was available.

| Split | # Domains (camera traps) | # Labeled examples | # Unlabeled examples |
|---|---|---|---|
| Source | 243 | 129,809 | 0 |
| Validation (ID) | | 7,314 | 0 |
| Target (ID) | | 8,154 | 0 |
| Validation (OOD) | 32 | 14,961 | 0 |
| Target (OOD) | 48 | 42,791 | 0 |
| Extra (OOD) | 3215 | 0 | 819,120 |
| Total | 3538 | 203,029 | 819,120 |

Table 3: Data for IWILDCAM2020-WILDS. Each domain corresponds to a different camera trap.

---

[3]The WCS Camera Traps Dataset can be found at http://lila.science/datasets/wcscameratraps

**Broader context.** There are large volumes of unlabeled natural world data that have been collected in growing repositories such as iNaturalist (Nugent, 2018), Wildlife Insights (Ahumada et al., 2020), and GBIF (Robertson et al., 2014). This data includes images or video collected by remote sensors or community scientists, GPS track data from an-animal devices, aerial data from drones or satellites, underwater sonar, bioacoustics, and eDNA. Methods that can harness the wealth of information in unlabeled ecological data are well-posed to make significant breakthroughs in how we think about ecological and conservation-focused research. Natural-world and ecological benchmarks that provide unlabeled data include NEWT (Van Horn et al., 2021), investigating efficient task learning, and Semi-Supervised iNat (Su & Maji, 2021), which provides labeled data for only a subset of the taxonomic tree. Recent work has begun to adapt weakly-supervised and self-supervised approaches for these natural world settings, including probing the generality and efficacy of self-supervision (Cole et al., 2021), incorporating domain-relevant context into self-supervision (Pantazis et al., 2021), or leveraging weak supervision from alternative data modalities (Weinstein et al., 2019) or pre-trained, generic models (Weinstein et al., 2021; Beery et al., 2019). Active learning also plays a role here in seeking to adapt models efficiently to unlabeled data from novel regions with only a few targeted labels (Kellenberger et al., 2019; Norouzzadeh et al., 2021).

## A.2 CAMELYON17-WILDS

The CAMELYON17-WILDS dataset (Koh et al., 2021) was adapted from the Camelyon17 dataset (Bandi et al., 2018), which is a collection of whole-slide images (WSIs) of breast cancer metastases in lymph node sections from 5 hospitals in the Netherlands. The labels were obtained by asking expert pathologists to perform pixel-level annotations of each WSI, which is an expensive and painstaking process. In practice, unlabeled WSIs (i.e., WSIs without pixel-level annotations) are much easier to obtain. For example, only a fraction of the WSIs in the original Camelyon17 dataset (Bandi et al., 2018) were labeled; the other WSIs, which are taken from the same 5 hospitals, were provided without labels. In this work, we augment the CAMELYON17-WILDS dataset with unlabeled data from these WSIs.

**Problem setting.** The task is to classify whether a histological image patch contains any tumor tissue. We consider generalizing from a set of training hospitals to new hospitals at test time. The input $x$ corresponds to a 96×96 image patch extracted from an WSI of a lymph node section, the label $y$ is a binary indicator of whether the central 32×32 patch of the input contains any pixel that was annotated as a tumor in the WSI, and the domain $d$ identifies which hospital the patch came from. Each patch also includes metadata on which WSI it was extracted from, though we do not use this metadata for training or evaluation. Models are evaluated by their average accuracy on a class-balanced test dataset.

**Data.** All of the labeled and unlabeled data are taken from the Camelyon17 dataset (Bandi et al., 2018), which consists of WSIs from 5 hospitals (domains) in the Netherlands. We provide unlabeled data from same domains as the labeled CAMELYON17-WILDS dataset (no extra domains). The domains are split as follows:

1. **Source:** Hospitals 1, 2, and 3.
2. **Validation (OOD):** Hospital 4.
3. **Target (OOD):** Hospital 5.

CAMELYON17-WILDS also includes a Validation (ID) set which contains data from the training hospitals.

The CAMELYON17-WILDS dataset has a total of 455,954 labeled patches across these splits, derived from the 10 WSIs per hospital that have full pixel-level annotations. We augment the dataset with a total of 2,999,307 unlabeled patches, extracted from an additional 90 unlabeled WSIs per hospital. There is no overlap between the WSIs used for the labeled versus unlabeled data. To extract and process each patch, we followed the same data processing steps that were carried out for the labeled data in Koh et al. (2021).

Unlike the labeled patches, which were sampled in a class-balanced manner (i.e., half of the patches have positive labels), we sampled the unlabeled patches uniformly at random from the unlabeled

| Split | # Domains (hospitals) | # Labeled examples | # Unlabeled examples |
|---|---|---|---|
| Source | 3 | 302,436 | 1,799,247 |
| Validation (ID) | | 33,560 | 0 |
| Validation (OOD) | 1 | 34,904 | 600,030 |
| Target (OOD) | 1 | 85,054 | 600,030 |
| Total | 5 | 455,954 | 2,999,307 |

Table 4: Data for CAMELYON17-WILDS. Each domain corresponds to a different hospital.

WSIs. We sampled 6,667 patches per unlabeled WSI, with the single exception of one WSI which had only 5,824 valid patches, resulting in a total of 3,000,150 unlabeled patches (Table 4). While the labeled patches were sampled in a class-balanced manner, the underlying label distribution skews heavily negative (approximately 95% of the patches in a WSI are negative), so we expect the unlabeled patches to be similarly skewed in their label distribution.

**Broader context.** We focused on providing unlabeled data from the same hospitals (domains) as in the original labeled CAMELYON17-WILDS dataset. This unlabeled data from the training and test hospitals can be used to develop and evaluate methods for semi-supervised learning (Peikari et al., 2018; Akram et al., 2018; Lu et al., 2019; Shaw et al., 2020) and domain adaptation (Ren et al., 2018; Zhang et al., 2019a; Koohbanani et al., 2021), respectively. In practice, there is also a large amount of unlabeled data from different domains that is publicly available: for example, The Cancer Genome Atlas (TCGA) hosts tens of thousands of publicly-available slide images across a variety of cancer types and from many different hospitals (Weinstein et al., 2013). These large and diverse datasets need not even be directly relevant to the task at hand, e.g., one could pre-train a model on images for different types of cancer even if the goal were to develop a model for breast cancer. Recent work has started to explore the use of these large and diverse datasets for computational pathology applications (Ciga et al., 2020; Dehaene et al., 2020) and in other medical imaging applications (Azizi et al., 2021).

## A.3 FMoW-WILDS

The FMoW-WILDS dataset (Koh et al., 2021) was adapted from the FMoW dataset (Christie et al., 2018), which consists of global satellite images from 2002–2018, labeled with the functional purpose of the buildings or land in the image. The labels are collected by a process which combines map data with crowdsourced annotations (from a trusted crowd). In contrast, unlabeled satellite imagery is readily available across the globe. In this work, we augment the FMoW-WILDS dataset with unused satellite images that were part of the original FMoW dataset but not in the FMoW-WILDS dataset.

**Problem setting.** The task is to classify the building or land-use type of a satellite image. We consider generalizing from images before 2013 to after 2013, as well as considering the performance on the worst-case geographic region (Africa, the Americas, Oceania, Asia, or Europe). The input $x$ is an RGB satellite image ($224 \times 224$ pixels). The label $y$ is one of 62 building or land use categories. The domain $d$ represents both the year and the geographical region of the image. Each image also includes metadata on the location and time of the image, although we do not use these except for splitting the domains. Models are evaluated by their average and worst-region accuracies in the OOD timeframe.

**Data.** The labeled and unlabeled data are taken from the FMoW dataset (Christie et al., 2018). We provide unlabeled data from same domains as the labeled FMoW-WILDS dataset (no extra domains). The domains are as follows:

1. **Source:** Images from 2002–2013.

2. **Validation (OOD):** Images from 2013–2016.

| Split | # Domains (years $\times$ region) | # Labeled examples | # Unlabeled examples |
|---|---|---|---|
| Source | $11 \times 5$ | 76,863 | 11,948 |
| Validation (ID) | | 11,483 | 0 |
| Target (ID) | | 11,327 | 0 |
| Validation (OOD) | $3 \times 5$ | 19,915 | 155,313 |
| Target (OOD) | $2 \times 5$ | 22,108 | 173,208 |
| Total | $16 \times 5$ | 141,696 | 340,469 |

Table 5: Data for FMOW-WILDS. Each domain corresponds to a different year and geographical region.

3. **Target (OOD):** Images from 2016–2018.

All of these domains have disjoint locations. FMOW-WILDS also includes Validation (ID) and Target (ID) sets which contain data from the training domains of 2002–2013.

The FMOW-WILDS dataset has 141,696 labeled images across these splits. We augment the dataset with 340,469 unlabeled images. These images come from two sources:

1. We use a sequestered split of the dataset, which consists of new locations that are not in the original labeled FMOW-WILDS dataset; these unlabeled data are drawn from the same distribution as the labeled data.

2. For the unlabeled target and validation splits, we also add unlabeled data in their respective timeframes from the training set locations. While the unlabeled data from the Validation (OOD) and Target (OOD) domains can come from the same locations as the labeled training data, we note that none of the locations in the labeled Validation (OOD) or Target (OOD) data, which is used for evaluation, is shared with any of the unlabeled or labeled data used for training.

**Broader context.** We focus on providing unlabeled data from the years (domains) that were in the original FMOW-WILDS dataset. Prior works have used unlabeled satellite imagery for pre-training (Xie et al., 2016; Jean et al., 2016; Xie et al., 2021a; Reed et al., 2021), self-training (Xie et al., 2021a), and semi-supervised learning (Reed et al., 2021). Leveraging unlabeled satellite imagery is powerful since it is widely available and can reduce the frequency at which we need to re-collect labeled data.

## A.4 POVERTYMAP-WILDS

The POVERTYMAP-WILDS dataset (Koh et al., 2021) was adapted from Yeh et al. (2020). The dataset consists of satellite images from 23 African countries, labeled with a village-level real-valued asset wealth index (measure of wealth). The labels are collected by conducting a nationally representative survey, which requires sending workers into the field to ask each household a number of questions and can be very expensive. In contrast, unlabeled satellite imagery is readily available across the globe. In this work, we augment the POVERTYMAP-WILDS dataset with satellite images from the same LandSat satellite.

**Problem setting.** The task is to predict a real-valued asset wealth index from a satellite image. We consider generalizing across country borders (the dataset contains 5 different cross validation folds, each splitting the countries differently). The input $x$ is a multispectral LandSat satellite image with 8 channels (resized to $224 \times 224$ pixels). The output $y$ is a real-valued asset wealth index. The domain $d$ represents the country the image was taken in, as well as whether the image was taken at an urban or rural area. Each image also includes metadata on the location and time, although we do not make use of these except for defining the domains. Models are evaluated by the average Pearson correlation ($r$) across 5 folds, as well as the lower of the Pearson correlations on the urban or rural

| Split | # Domains (countries $\times$ rural-urban) | # Labeled ex. | # Unlabeled ex. |
|---|---|---|---|
| Source | $13 \times 2$ | 9,797 | 181,948 |
| Validation (ID) | | 1,000 | 0 |
| Target (ID) | | 1,000 | 0 |
| Validation (OOD) | $5 \times 2$ | 3,909 | 24,173 |
| Target (OOD) | $5 \times 2$ | 3,963 | 55,275 |
| Total | $23 \times 2$ | 19,669 | 261,396 |

Table 6: Data for POVERTYMAP-WILDS (Fold A). Each domain corresponds to a different country and whether the image was from a rural or urban area.

subpopulations to test generalization to these subpopulations. In particular, generalization to rural subpopulations is important as poverty is more common in rural areas.

**Data.**  We provide unlabeled data from same domains as the labeled POVERTYMAP-WILDS dataset (no extra domains). The domains are split as follows:

1. **Source:** Images from training countries in the fold.

2. **Validation (OOD):** Images from validation countries in the fold.

3. **Target (OOD):** Images from test countries in the fold.

All the countries in these splits are disjoint. Folds also contain a Validation (ID) and Target (ID) set with data from the training countries.

The POVERTYMAP-WILDS dataset has 19,669 labeled images across these splits. We augment the dataset with 261,396 unlabeled images from the same 23 countries. These images are collected using the same process as Yeh et al. (2020) from the same LandSat satellite. The image locations are chosen to be roughly near survey locations from the Demographic and Health Surveys (DHS).

**Broader context.**  We focus on providing unlabeled data from the countries (domains) that were in the original POVERTYMAP-WILDS dataset. Prior works on poverty prediction have used pre-training on unlabeled data (to predict an auxiliary task such as nighttime light prediction) (Xie et al., 2016; Jean et al., 2016) and for semi-supervised learning via entropy minimization (Jean et al., 2018). However, these works focus on generalization to new locations in the countries in the training set. Poverty prediction is different from usual tasks in that the output is real-valued. Most methods for unlabeled data are made for classification tasks, and we hope that our dataset will encourage more work on methods for using unlabeled data for improving OOD performance in regression tasks.

## A.5   GLOBALWHEAT-WILDS

The GLOBALWHEAT-WILDS dataset was extended from the Global Wheat Head Dataset developed by David et al. (2020; 2021). The goal of the dataset is to localize wheat heads from field images to assist plant scientists to assess the density, size, and health of wheat heads in a particular wheat field. This imagery is acquired during different periods to cover the development of the vegetation, from the emergence to organ appearance. Examples in GLOBALWHEAT-WILDS are labeled by bounding box annotations of each wheat head in the image. Wheat heads are densely packed and overlapping, making object annotation highly tedious. Thus, the Global Wheat Head Dataset (GWHD) is relatively small, while in reality more field images are available. We supplement GLOBALWHEAT-WILDS with unlabeled examples from the same set of field vehicles and sensors but taken in different acquisition sessions, i.e., at different locations or the same location in a different year. The inclusion of this unlabeled data allows: 1) a much higher spatial coverage of a field location when the data comes from an acquisition session which is already included, 2) a much higher temporal resolution when the data comes from a location which is already included, so we have a larger range of wheat

| Split | # Domains (acquisition session) | # Labeled examples | # Unlabeled examples |
|---|---|---|---|
| Source | | 2,943 | 5,997 |
| Validation (ID) | 18 | 357 | 0 |
| Target (ID) | | 357 | 0 |
| Validation (OOD) | 8 | 1,424 | 2,000 |
| Target (OOD) | 21 | 1,434 | 8,997 |
| Extra | 53 | 0 | 42,445 |
| Total | 100 | 6,515 | 59,439 |

Table 7: Data for GLOBALWHEAT-WILDS.

growth stages, and 3) slightly more diversity when the session comes from a different location, but with the same image acquisition protocol (i.e., the same field vehicle and image sensor).

**Problem setting.** The task is to localize wheat heads in high resolution overhead field images taken from above the crop canopy. We consider generalizing across acquisition sessions representing a particular location, time and sensor with which the images were captured. Variation across sessions includes changes in wheat genotype, wheat head appearance, growing conditions, background appearance, illumination and acquisition protocol. The input $x$ is an overhead outdoor image of wheat canopy, and the label $y$ is a set of box coordinates bounding the wheat heads (the spike at the top of the wheat plant holding grain), omitting any hair-like awns that may extend from the head. The domain $d$ designates an acquisition session, which corresponds to a certain location, time, and imaging sensor.

**Data.** We provide unlabeled data from same domains as the labeled GLOBALWHEAT-WILDS dataset. Additionally, we provide unlabeled data from extra acquisition sessions not in the labeled GLOBALWHEAT-WILDS dataset (extra domains). The domains are split as follows:

1. **Source:** 18 acquisition sessions in Europe (France ×13, Norway ×2, Switzerland, United Kingdom, Belgium).

2. **Validation (OOD):** 8 acquisition sessions: 7 in Asia (Japan × 4, China × 3) and 1 in Africa (Sudan).

3. **Target (OOD):** 21 acquisition sessions: 11 in Australia and 10 in North America (USA × 6, Mexico × 3, Canada).

4. **Extra (OOD):** 53 acquisition sessions distributed across the world.

The source, validation, and target sessions are split by continent, while the extra sessions are taken from across the world. For acquisition sessions with both labeled and unlabeled data, we randomly selected new patches of 1024x1024 pixels from the original underlying data. The images were preprocessed in the same way as described in David et al. (2021).

**Broader context.** Utilizing unlabeled data is relatively new in the context of plant phenotyping, due to the lack of a large, unlabeled database of plant images. However, larger plant image datasets are starting become available, such as from the Terraphenotying Reference Platform (TERRA-Ref, Burnette et al. (2018)). Increasing the sample size and variation within plant datasets is an important goal, because plants from the same species are fairly self-similar within the same field and therefore increasing the number of locations, times and image types included in a dataset can be beneficial for making fine-grained visual classifications for plants. Further, for plant phenotyping to be used in farming applications, such as for precisely spraying weeds in a field with herbicide, models must be highly robust to variations between different fields.

| Split | Name | Country | Site | Date | Sensor | Stage | #Labeled | #Heads | #Unlabeled |
|---|---|---|---|---|---|---|---|---|---|
| Source | Arvalis_1 | France | Gréoux | 6/2/2018 | Handheld | PF | 66 | 2935 | 0 |
| Source | Arvalis_2 | France | Gréoux | 6/16/2018 | Handheld | F | 401 | 21003 | 0 |
| Source | Arvalis_3 | France | Gréoux | 7/1/2018 | Handheld | F-R | 588 | 21893 | 0 |
| Source | Arvalis_4 | France | Gréoux | 5/27/2019 | Handheld | F | 204 | 4270 | 0 |
| Source | Arvalis_5 | France | VLB* | 6/6/2019 | Handheld | F | 448 | 8180 | 0 |
| Source | Arvalis_6 | France | VSC* | 6/26/2019 | Handheld | F-R | 160 | 8698 | 0 |
| Source | Arvalis_7 | France | VLB* | 6/1/2019 | Handheld | F-R | 24 | 1247 | 0 |
| Source | Arvalis_8 | France | VLB* | 6/1/2019 | Handheld | F-R | 20 | 1062 | 0 |
| Source | Arvalis_9 | France | VLB* | 6/1/2020 | Handheld | R | 32 | 1894 | 0 |
| Source | Arvalis_10 | France | Mons | 6/10/2020 | Handheld | F | 60 | 1563 | 1000 |
| Source | Arvalis_11 | France | VLB* | 6/18/2020 | Handheld | F | 60 | 2818 | 0 |
| Source | Arvalis_12 | France | Gréoux | 6/15/2020 | Handheld | F | 29 | 1277 | 1000 |
| Source | ETHZ_1 | Switzerland | Eschikon | 6/6/2018 | Spidercam | F | 747 | 49603 | 0 |
| Source | INRAE_1 | France | Toulouse | 5/28/2019 | Handheld | F-R | 176 | 3634 | 1000 |
| Source | NMBU_1 | Norway | NMBU | 7/24/2020 | Cart | F | 82 | 7345 | 999 |
| Source | NMBU_2 | Norway | NMBU | 8/7/2020 | Cart | R | 98 | 5211 | 998 |
| Source | Rres_1 | UK | Rothamsted | 7/13/2015 | Gantry | F-R | 432 | 19210 | 0 |
| Source | ULiège_1 | Belgium | Gembloux | 7/28/2020 | Cart | R | 30 | 1847 | 1000 |
| Validation | ARC_1 | Sudan | WadMedani | 3/1/2021 | Handheld | F | 30 | 1169 | 0 |
| Validation | NAU_1 | China | Baima | n/a | Handheld | PF | 20 | 1240 | 0 |
| Validation | NAU_2 | China | Baima | 5/2/2020 | Cart | PF | 100 | 4918 | 1000 |
| Validation | NAU_3 | China | Baima | 5/9/2020 | Cart | F | 100 | 4596 | 1000 |
| Validation | Ukyoto_1 | Japan | Kyoto | 4/30/2020 | Handheld | PF | 60 | 2670 | 0 |
| Validation | Utokyo_1 | Japan | Tsukuba | 5/22/2018 | Cart | R | 538 | 14185 | 0 |
| Validation | Utokyo_2 | Japan | Tsukuba | 5/22/2018 | Cart | R | 456 | 13010 | 0 |
| Validation | Utokyo_3 | Japan | Hokkaido | 6/16/2021 | Handheld | multiple | 120 | 3085 | 0 |
| Target | CIMMYT_1 | Mexico | CiudadObregon | 3/24/2020 | Cart | PF | 69 | 2843 | 1000 |
| Target | CIMMYT_2 | Mexico | CiudadObregon | 3/19/2020 | Cart | PF | 77 | 2771 | 1000 |
| Target | CIMMYT_3 | Mexico | CiudadObregon | 3/23/2020 | Cart | PF | 60 | 1561 | 1000 |
| Target | KSU_1 | US | Manhattan,KS | 5/19/2016 | Tractor | PF | 100 | 6435 | 1000 |
| Target | KSU_2 | US | Manhattan,KS | 5/12/2017 | Tractor | PF | 100 | 5302 | 1000 |
| Target | KSU_3 | US | Manhattan,KS | 5/25/2017 | Tractor | F | 95 | 5217 | 1000 |
| Target | KSU_4 | US | Manhattan,KS | 5/25/2017 | Tractor | R | 60 | 3285 | 1000 |
| Target | Terraref_1 | US | Maricopa | 4/2/2020 | Gantry | R | 144 | 3360 | 997 |
| Target | Terraref_2 | US | Maricopa | 3/20/2020 | Gantry | F | 106 | 1274 | 1000 |
| Target | UQ_1 | Australia | Gatton | 8/12/2015 | Tractor | PF | 22 | 640 | 0 |
| Target | UQ_2 | Australia | Gatton | 9/8/2015 | Tractor | PF | 16 | 39 | 0 |
| Target | UQ_3 | Australia | Gatton | 9/15/2015 | Tractor | F | 14 | 297 | 0 |
| Target | UQ_4 | Australia | Gatton | 10/1/2015 | Tractor | F | 30 | 1039 | 0 |
| Target | UQ_5 | Australia | Gatton | 10/9/2015 | Tractor | F-R | 30 | 3680 | 0 |
| Target | UQ_6 | Australia | Gatton | 10/14/2015 | Tractor | F-R | 30 | 1147 | 0 |
| Target | UQ_7 | Australia | Gatton | 10/6/2020 | Handheld | R | 17 | 1335 | 0 |
| Target | UQ_8 | Australia | McAllister | 10/9/2020 | Handheld | R | 41 | 4835 | 0 |
| Target | UQ_9 | Australia | Brookstead | 10/16/2020 | Handheld | F-R | 33 | 2886 | 0 |
| Target | UQ_10 | Australia | Gatton | 9/22/2020 | Handheld | F-R | 106 | 8629 | 0 |
| Target | UQ_11 | Australia | Gatton | 8/31/2020 | Handheld | PF | 84 | 4345 | 0 |
| Target | Usask_1 | Canada | Saskatoon | 6/6/2018 | Tractor | F-R | 200 | 5985 | 0 |

Table 8: Source, validation, and test domains for GLOBALWHEAT-WILDS.

| Split | Name | Country | Site | Date | Sensor | Stage | #Labeled | #Heads | #Unlabeled |
|---|---|---|---|---|---|---|---|---|---|
| Extra | Arvalis_13 | France | Mons | 6/15/2018 | Handheld | F-R | 0 | 0 | 995 |
| Extra | Arvalis_14 | France | Gréoux | 5/25/2020 | Handheld | F | 0 | 0 | 1000 |
| Extra | Arvalis_15 | France | VLB* | 6/2/2020 | Handheld | F | 0 | 0 | 1000 |
| Extra | Arvalis_16 | France | Gréoux | 6/22/2020 | Handheld | F-R | 0 | 0 | 1000 |
| Extra | Arvalis_17 | France | Bignan | 5/18/2021 | Handheld | F-R | 0 | 0 | 1000 |
| Extra | Arvalis_18 | France | VLB* | 5/28/2021 | Handheld | PF | 0 | 0 | 1000 |
| Extra | Arvalis_19 | France | Encrambade | 6/2/2021 | Handheld | F | 0 | 0 | 1000 |
| Extra | Arvalis_20 | France | OLM* | 6/2/2021 | Handheld | F | 0 | 0 | 1000 |
| Extra | Arvalis_21 | France | Encrambade | 6/11/2021 | Handheld | PF | 0 | 0 | 1000 |
| Extra | Arvalis_22 | France | VLB* | 6/14/2021 | Handheld | F | 0 | 0 | 1000 |
| Extra | Arvalis_23 | France | OLM* | 6/17/2021 | Handheld | F-R | 0 | 0 | 1000 |
| Extra | CIMMYT_4 | Mexico | CiudadObregon | 3/11/2020 | Cart | F | 0 | 0 | 1000 |
| Extra | CIMMYT_5 | Mexico | CiudadObregon | 3/12/2020 | Cart | F | 0 | 0 | 1000 |
| Extra | CIMMYT_6 | Mexico | CiudadObregon | 3/13/2020 | Cart | F | 0 | 0 | 1000 |
| Extra | CIMMYT_7 | Mexico | CiudadObregon | 3/13/2020 | Cart | F | 0 | 0 | 1000 |
| Extra | CIMMYT_8 | Mexico | CiudadObregon | 3/13/2020 | Cart | F | 0 | 0 | 1000 |
| Extra | CIMMYT_9 | Mexico | CiudadObregon | 3/19/2020 | Cart | F | 0 | 0 | 1000 |
| Extra | CIMMYT_10 | Mexico | CiudadObregon | 4/15/2020 | Cart | E | 0 | 0 | 1000 |
| Extra | CIMMYT_11 | Mexico | CiudadObregon | 4/22/2020 | Cart | E | 0 | 0 | 1000 |
| Extra | CIMMYT_12 | Mexico | CiudadObregon | 4/22/2020 | Cart | E | 0 | 0 | 1000 |
| Extra | CIMMYT_13 | Mexico | CiudadObregon | 4/22/2020 | Cart | E | 0 | 0 | 1000 |
| Extra | CIMMYT_14 | Mexico | CiudadObregon | 4/22/2020 | Cart | PF | 0 | 0 | 1000 |
| Extra | CIMMYT_15 | Mexico | CiudadObregon | 4/28/2020 | Cart | PF | 0 | 0 | 1000 |
| Extra | CIMMYT_16 | Mexico | CiudadObregon | 5/3/2020 | Cart | F-R | 0 | 0 | 1000 |
| Extra | ETHZ_2 | Switzerland | Eschikon | 6/6/2018 | Spidercam | F | 0 | 0 | 750 |
| Extra | INRAE_2 | France | Clermont-Ferrand | 5/29/2019 | Handheld | F | 0 | 0 | 1000 |
| Extra | KSU_5 | US | Manhattan,KS | 5/4/2016 | Tractor | F | 0 | 0 | 1000 |
| Extra | KSU_6 | US | Manhattan,KS | 4/23/2017 | Tractor | P-F | 0 | 0 | 1000 |
| Extra | Rres_2 | UK | Rothamsted | 7/7/2015 | Gantry | R | 0 | 0 | 1000 |
| Extra | Rres_3 | UK | Rothamsted | 7/10/2015 | Gantry | F | 0 | 0 | 1000 |
| Extra | Rres_4 | UK | Rothamsted | 7/13/2015 | Gantry | F-R | 0 | 0 | 1000 |
| Extra | Rres_5 | UK | Rothamsted | 7/20/2015 | Gantry | F-R | 0 | 0 | 1000 |
| Extra | ULiège_2 | Belgium | Gembloux | 6/11/2020 | Cart | PF | 0 | 0 | 1000 |
| Extra | ULiège_3 | Belgium | Gembloux | 6/15/2020 | Cart | F | 0 | 0 | 1000 |
| Extra | ULiège_4 | Belgium | Gembloux | 6/16/2020 | Cart | F | 0 | 0 | 1000 |
| Extra | ULiège_5 | Belgium | Gembloux | 6/18/2020 | Cart | F | 0 | 0 | 1000 |
| Extra | ULiège_6 | Belgium | Gembloux | 6/23/2020 | Cart | F | 0 | 0 | 1000 |
| Extra | ULiège_7 | Belgium | Gembloux | 6/26/2020 | Cart | F | 0 | 0 | 1000 |
| Extra | ULiège_8 | Belgium | Gembloux | 7/7/2020 | Cart | F-R | 0 | 0 | 1000 |
| Extra | ULiège_9 | Belgium | Gembloux | 7/13/2020 | Cart | F-R | 0 | 0 | 1000 |
| Extra | Usask_2 | Canada | Saskatchewan | 8/6/2019 | Tractor | F | 0 | 0 | 800 |
| Extra | Usask_3 | Canada | Saskatchewan | 8/12/2019 | Tractor | F-R | 0 | 0 | 800 |
| Extra | Utokyo_4 | Japan | Hokkaido | 6/7/2021 | Handheld | PF | 0 | 0 | 100 |
| Extra | Utokyo_5 | Japan | Hokkaido | 6/9/2021 | Handheld | F | 0 | 0 | 100 |
| Extra | Utokyo_6 | Japan | Hokkaido | 6/16/2021 | Handheld | PF | 0 | 0 | 100 |
| Extra | Utokyo_7 | Japan | Hokkaido | 6/23/2021 | Handheld | F | 0 | 0 | 100 |
| Extra | Utokyo_8 | Japan | Hokkaido | 7/3/2021 | Handheld | F | 0 | 0 | 100 |
| Extra | Utokyo_9 | Japan | Hokkaido | 7/10/2021 | Handheld | F | 0 | 0 | 100 |
| Extra | Utokyo_10 | Japan | Hokkaido | 7/10/2021 | Handheld | F-R | 0 | 0 | 100 |
| Extra | Utokyo_11 | Japan | Hokkaido | 7/11/2021 | Handheld | F-R | 0 | 0 | 100 |
| Extra | Utokyo_12 | Japan | Hokkaido | 7/20/2021 | Handheld | R | 0 | 0 | 100 |
| Extra | Utokyo_13 | Japan | Hokkaido | 7/20/2021 | Handheld | R | 0 | 0 | 100 |
| Extra | Utokyo_14 | Japan | Hokkaido | 7/28/2021 | Handheld | R | 0 | 0 | 100 |

Table 9: Extra domains for GLOBALWHEAT-WILDS.

| Split | # Domains (scaffolds) | # Labeled examples | # Unlabeled examples |
|---|---|---|---|
| Source | 44,930 | 350,343 | 4,052,627 |
| Validation (OOD) | 31,361 | 43,793 | 430,325 |
| Target (OOD) | 43,793 | 43,793 | 517,048 |
| Total | 120,084 | 437,929 | 5,000,000 |

Table 10: Data for OGB-MOLPCBA. Each domain corresponds to a different molecule scaffold structure.

## A.6 OGB-MOLPCBA

The OGB-MOLPCBA dataset was adapted from the Open Graph Benchmark (Hu et al., 2020b) and originally curated by the MoleculeNet (Wu et al., 2018) from the PubChem database (Bolton et al., 2008). The dataset is a collection of molecules annotated with 128 kinds of binary labels indicating the outcome of different biological assays. Performing biological assays is expensive, and as a result, the assay labels are only sparsely available over a tiny portion of the molecules curated in the large-scale PubChem database (Bolton et al., 2008). On the other hand, unlabeled molecule data is abundant and readily available from the database. Prior work in graph machine learning has leveraged unlabeled molecules to perform pre-training (Hu et al., 2020c) and semi-supervised learning (Sun et al., 2020). In this work, we augment the OGB-MOLPCBA dataset with unlabeled molecules subsampled from the PubChem database.

**Problem setting.** The task is multi-task molecule classification, and we consider generalizing to new molecule scaffold structures at test time. The input $x$ corresponds to a molecular graph (where nodes are atoms and edges are chemical bounds), the label $y$ is a 128-dimensional binary vector, representing the binary outcomes of the biological assay results. $y$ could contain NaN values, indicating that the corresponding biological assays were not performed on the given molecule. The domain $d$ indicates the scaffold group a molecule belongs to. As the binary labels are highly-skewed, the model's classification performance is evaluated using the Average Precision.

**Data.** All of the labeled and unlabeled data are taken from the PubChem database (Bolton et al., 2008). We provide unlabeled data from same domains as the labeled OGB-MOLPCBA dataset (no extra domains). We curate the unlabeled data by randomly sampling 5 million molecules from the PubChem database. We then assign these unlabeled molecules to the existing labeled scaffold groups that contain the most similar molecules. Specifically, we first compute the 1024-dimensional Morgan fingerprints for all the molecules (Rogers & Hahn, 2010; Landrum et al., 2006). Then, for each unlabeled molecule, we compute its Jaccard similarity against all the labeled molecules in OGB-MOLPCBA and obtain a labeled molecule with the highest Jaccard similarity. Finally, we assign the unlabeled molecule to the scaffold group that the most similar labeled molecule belongs to. This way, the molecules within the same scaffold groups are structurally similar to each other.

The domains in the OGB-MOLPCBA dataset are as follows:

1. **Source:** 44,930 scaffold groups.

2. **Validation (OOD):** 31,361 scaffold groups.

3. **Target (OOD):** 43,793 scaffold groups.

The largest scaffolds are in the source split and the smallest scaffolds in the target split. We assign all of the unlabeled molecules to the existing domains, so there are no extra domains added.

While the unlabeled data are similar to the labeled data in that they were all derived from PubChem (Bolton et al., 2008), it is quite possible that there was some selection bias in which molecules in PubChem were chosen to be labeled, which would lead to an undocumented distribution shift between the unlabeled and labeled datasets.

**Broader context.** We focused on providing unlabeled data for both training and OOD test domains. Unlabeled molecules can be used to develop and evaluate methods for domain adaptation, self-training, as well as pre-training (Hu et al., 2020c) and semi-supervised learning (Sun et al., 2020). In terms of future directions, we think it is fruitful to explore both graph-agnostic methods (e.g., pseudo-label training) and more graph-specific methods (e.g., self-supervised learning of graph neural networks (Xie et al., 2021b)).

### A.7 CIVILCOMMENTS-WILDS

The CIVILCOMMENTS-WILDS dataset (Koh et al., 2021) was adapted from the CivilComments dataset (Borkan et al., 2019), which is a collection of text comments made on online articles. The data in CIVILCOMMENTS-WILDS underwent a significant labeling and annotation process: each example was labeled toxic or non-toxic and annotated for whether they mentioned certain demographic identities by at least 10 crowdworkers. Such a substantial labeling and identity annotation process is expensive and time-consuming. On the other hand, unlabeled, unannotated text comments are readily available. For example, CIVILCOMMENTS-WILDS only contains a subset of all data available in the original CivilComments dataset (Borkan et al., 2019), most of which Koh et al. (2021) excluded because these examples were not annotated for mentioning identities. In this work, we augment the CIVILCOMMENTS-WILDS dataset with these unlabeled, unannotated comments.

**Problem setting.** The task is to classify whether a text comment is toxic or not. The input $x$ is a text comment (at least one sentence long) originally made on an online article, the label $y$ is a binary indicator of whether the comment is rated toxic or not, and the domain $d$ is an 8-dimensional binary vector, where each dimension corresponds to whether the comment mentions each of 8 demographic identities: *male, female, LGBTQ, Christian, Muslim, other religions, Black*, or *White*, respectively. Each comment also includes metadata on which article the comment was made on, although we do not use this metadata for training or evaluation.

We consider the subpopulation shift setting, where the model must perform well across all subpopulations, which are defined based on $d$. Koh et al. (2021) define 16 subpopulations (groups) based on $d$. Models are then evaluated by their worst-group accuracy, i.e., the lowest accuracy over the 16 groups considered. In our work, we use the same evaluation setup.

**Data.** All of the labeled and unlabeled data are taken from the CivilComments dataset (Borkan et al., 2019). After preprocessing, Koh et al. (2021) created the CIVILCOMMENTS-WILDS dataset using the 448,000 examples that were fully annotated for both toxicity $y$ and the mention of demographic identities $d$. In this work, we augment CIVILCOMMENTS-WILDS with an additional 1,551,515 examples collected by Borkan et al. (2019). We use these examples as unlabeled data. We follow the same preprocessing steps as was done with the labeled data in Koh et al. (2021). The resulting unlabeled examples have no identity annotations $d$ and no toxicity label $y$. We note that Borkan et al. (2019) actually do provide toxicity labels for these examples in the original CivilComments dataset, but we ignore these labels and use them neither for training nor evaluation.

Because our unlabeled examples have no identity annotations, we cannot group these examples as Koh et al. (2021) group the labeled examples; thus we refer to this data as unlabeled data coming from extra domains (Table 11). In practice, these comments may actually mention any number of identities.

A substantial amount (1,427,848 or 92%) of the unlabeled comments are drawn from the same articles as the labeled comments. In particular, 140,082 unlabeled comments are from the same articles as labeled comments in the test split.

CIVILCOMMENTS-WILDS exhibits class imbalance. We account for this when benchmarking methods by sampling class-balanced batches of labeled data when applicable (see Appendix B).

**Broader context.** In this work, we focused on supplementing CIVILCOMMENTS-WILDS with extra unannotated data from the original CivilComments dataset (Borkan et al., 2019). In practice, unannotated text comments are widely available on the internet. Whether using such unlabeled data, as we do in this work, can help with bias is still an open question. Previous work suggests that training on large amounts of data alone is not sufficient to avoid unwanted biases, since many

| Split | # Domains (label × identity groups) | # Labeled examples | # Unlabeled examples |
|---|---|---|---|
| Source | 16 | 269,038 | 0 |
| Validation | 16 | 45,180 | 0 |
| Target | 16 | 133,782 | 0 |
| Extra | 1 | 0 | 1,551,515 |
| Total | 16 | 448,000 | 1,551,515 |

Table 11: Data for CIVILCOMMENTS-WILDS. All of the splits are identically distributed.

papers have pointed out biases in large language models (Abid et al., 2021; Nadeem et al., 2020; Gehman et al., 2020). However, recent work has also suggested that pre-trained models can be trained to be more robust against some types of spurious correlations (Hendrycks et al., 2020; Tu et al., 2020) and that additional domain- and task-specific pre-training (Gururangan et al., 2020) can also improve performance. We hope our contributions to the CIVILCOMMENTS-WILDS dataset can encourage future study on whether unlabeled data can be leveraged to improve generalization across subpopulation shifts.

## A.8    AMAZON-WILDS

The AMAZON-WILDS dataset (Koh et al., 2021) was adapted from the Amazon reviews dataset (Ni et al., 2019), which is a collection of product reviews written by reviewers. While Amazon reviews are always labeled by the star ratings in practice, unlabeled data is a common source of leverage more generally for sentiment classification, with prior work in domain adaptation (Blitzer & Pereira, 2007; Glorot et al., 2011) and semi-supervised learning (Dasgupta & Ng, 2009; Li et al., 2011). In this work, we augment the AMAZON-WILDS dataset with unlabeled reviews, whose star ratings have been removed.

**Problem setting.**    The task is sentiment classification, and we consider generalizing from a set of reviewers to new reviewers at test time. The input $x$ corresponds to a review text, the label $y$ is the star rating from 1 to 5, and the domain $d$ identifies which user wrote the review. For each review, additional metadata (product ID, product category, review time, and summary) are also available. Because the goal is to train a model that performs well across a wide range of reviewers, models are evaluated by their tail performance, concretely, their accuracy on the user at the 10th percentile.

**Data.**    All of the labeled and unlabeled data are taken from the Amazon reviews dataset (Ni et al., 2019). We provide unlabeled data from same domains as the labeled AMAZON-WILDS dataset. Additionally, we provide unlabeled data from extra reviewers not in the labeled AMAZON-WILDS dataset (extra domains). The domains are split as follows:

1. **Source:** 1,252 reviewers.

2. **Validation (OOD):** 1,334 reviewers.

3. **Target (OOD):** 1,334 reviewers.

4. **Extra (OOD):** 21,694 reviewers.

The reviewers in each split are distinct, and all reviewers have at least 75 reviews. The distributions of reviewers in each split are identical. AMAZON-WILDS also includes Validation (ID) and Target (ID) sets which contain data from the source reviewers.

The AMAZON-WILDS dataset has a total of 539,502 labeled reviews across these splits, and we augment the dataset with a total of 3,462,668 unlabeled reviews. For each split of the unlabeled data, we include all available reviews that are written by the reviewer. For the Extra (OOD) split, we include all reviewers with at least 75 reviews that are not in Source, Validation (OOD), or Target (OOD) splits.

| Split | # Domains (reviewers) | # Labeled examples | # Unlabeled examples |
|---|---|---|---|
| Source | | 245,502 | 0 |
| Validation (ID) | 1,252 | 46,950 | 0 |
| Target (ID) | | 46,950 | 0 |
| Validation (OOD) | 1,334 | 100,050 | 266,066 |
| Target (OOD) | 1,334 | 100,050 | 268,761 |
| Extra (OOD) | 21,694 | 0 | 2,927,841 |
| Total | 25,614 | 539,502 | 3,462,668 |

Table 12: Data for AMAZON-WILDS. Each domain corresponds to a different reviewer.

To filter and process reviews, we followed the same data processing steps as for the labeled data in AMAZON-WILDS (Koh et al., 2021).

**Broader context.** We focused on providing unlabeled data from OOD domains, including both test and extra domains. Unlabeled data from the test reviewers can be used to develop and evaluate methods for domain adaptation (Ren et al., 2018; Zhang et al., 2019a; Koohbanani et al., 2021), which has been well-studied in the context of sentiment classification (Blitzer & Pereira, 2007; Glorot et al., 2011). While there is limited prior work on leveraging unlabeled data from extra domains, some domain adaptation techniques can be readily adapted to leverage such unlabeled data (Ganin et al., 2016). Finally, we focus on unlabeled data specific to the task in this work, varying only the domains, and this contrasts with the type of unlabeled data used for pre-training in NLP, which is much larger and more diverse (Devlin et al., 2019; Brown et al., 2020).

# B    ALGORITHM DETAILS

## B.1    EMPIRICAL RISK MINIMIZATION (ERM)

As a baseline, we consider Empirical Risk Minimization (ERM). ERM ignores unlabeled data and minimizes the average labeled loss. We additionally evaluate ERM with strong data augmentation on applicable datasets, i.e., on IWILDCAM2020-WILDS, CAMELYON17-WILDS, POVERTYMAP-WILDS, and FMOW-WILDS (see Appendix C). ERM with strong data augmentation learns a model $h$ that minimizes the labeled training loss

$$L_{\mathrm{L}}(h) = \frac{1}{n_{\mathrm{L}}} \sum_{i=1}^{n_{\mathrm{L}}} \ell\big(h \circ A_{\mathrm{strong}}(x_{\mathrm{L}}^{(i)}), y_{\mathrm{L}}^{(i)}\big), \tag{1}$$

where $A_{\mathrm{strong}}$ is a stochastic data augmentation operation, and $\ell$ measures the prediction loss. We use $L_{\mathrm{L}}$ throughout this appendix to refer to the above labeled loss *with* strong augmentations (on applicable datasets).

For all dataset except CIVILCOMMENTS-WILDS, we sample labeled batches uniformly at random. In our experiments, we account for class imbalance in CIVILCOMMENTS-WILDS by explicitly sampling class-balanced batches of labeled data when computing $L_{\mathrm{L}}(h)$.

## B.2    DOMAIN-INVARIANT METHODS

Domain-invariant methods seek to learn feature representations that are invariant across domains. These methods are motivated by earlier theoretical results showing that the gap between in- and out-of-distribution performance depends on some measure of divergence between the source and target distributions (Ben-David et al., 2010). To minimize this divergence, the methods described below penalize divergence between feature representations across domains, i.e., they encourage the model to produce feature representations that are similar across domains.

Consider a model $h = g \circ f$, where the featurizer $f : \mathcal{X} \to \mathcal{F}$ maps the inputs to some feature space, and the head $g : \mathcal{F} \to \mathcal{Y}$ maps feature representations to prediction targets. Domain-invariant methods seek to constrain $f$ to output similar representations for labeled and unlabeled data.

In this work, we adapt all of our domain-invariant methods to use data augmentations on applicable datasets (see Appendix C), and thus the output of $f$ on the labeled batch is

$$B_{\mathrm{L}} = \{f \circ A_{\mathrm{strong}}(x_{\mathrm{L}}^{(i)}) : i \in (1, \cdots, n_{\mathrm{L}})\} \tag{2}$$

Similarly, the output of $f$ on an unlabeled batch is

$$B_{\mathrm{U}} = \{f \circ A_{\mathrm{strong}}(x_{\mathrm{U}}^{(i)}) : i \in (1, \cdots, n_{\mathrm{U}})\} \tag{3}$$

Domain-invariant methods seek to minimize some divergence $\xi : \mathcal{F} \times \mathcal{F} \to \mathbb{R}$ between the labeled data $B_{\mathrm{L}}$ and the unlabeled data $B_{\mathrm{U}}$, where the choice of divergence depends on the specific method. The divergence is expressed as a penalty term:

$$L_{\mathrm{penalty}}(f) = \xi\Big(B_{\mathrm{L}}, B_{\mathrm{U}}\Big) \tag{4}$$

The final objective is a combination of the labeled loss and penalty loss. The balance between the two losses is controlled by hyperparameter $\lambda$, the penalty weight.

$$L(h) = L_{\mathrm{L}}(h) + \lambda L_{\mathrm{penalty}}(f) \tag{5}$$

In our experiments, we study two classical domain-invariant methods, Correlation Alignment (CORAL) (Sun et al., 2016; Sun & Saenko, 2016) and Domain-Adversarial Neural Networks (DANN) (Ganin et al., 2016). These methods are well-known and established, but their performance can be lower than that of newer domain-invariant methods that employ different penalties to encourage the source and target representations to be similar (Jiang et al., 2020; Zhang et al., 2021). Examples of these newer methods are Joint Adaptation Networks (JAN) (Long et al., 2017), Conditional Domain Adversarial Networks (CDAN) (Long et al., 2018), Collaborative and Adversarial

Networks (CAN) (Zhang et al., 2018), and models with Adaptive Feature Norm (AFN) (Xu et al., 2019), as well as methods that minimize the Maximum Classifier Discrepancy (MCD) (Saito et al., 2018) and the Margin Disparity Discrepancy (MDD) (Zhang et al., 2019b).

All of the above methods were developed for the single-source single-target setting, where the source domain is treated as a single distribution, and likewise for the target domain. As each WILDS 2.0 dataset comprises multiple source domains and multiple target domains, it is likely that methods that can leverage this additional structure could perform better. Examples of these methods include Multi-source Domain Adversarial Networks (MDAN) (Zhao et al., 2018) and Moment Matching for Multi-Source Domain Adaptation (M3SDA) (Peng et al., 2019). The DomainBed (Gulrajani & Lopez-Paz, 2020) and WILDS (Koh et al., 2021) benchmarks also extended single-source algorithms like CORAL and DANN to take advantage of multiple source domains in the domain generalization setting, and similar extensions in the domain adaptation setting could be promising.

**Correlation Alignment (CORAL).** Algorithm 1 describes CORAL, proposed by Sun et al. (2016); Sun & Saenko (2016). CORAL measures the divergence $\xi$ between batches of feature representations in terms of the deviation between their first and second order statistics. Given a labeled batch and unlabeled batch of features $B_{\mathrm{L}} \in \mathbb{R}^{n_{\mathrm{L}} \times m}, B_{\mathrm{U}} \in \mathbb{R}^{n_{\mathrm{U}} \times m}$, define the feature means as

$$\mu_{\mathrm{L}} = \frac{1}{n_{\mathrm{L}}} 1^T B_{\mathrm{L}} \tag{6}$$

$$\mu_{\mathrm{U}} = \frac{1}{n_{\mathrm{U}}} 1^T B_{\mathrm{U}} \tag{7}$$

and covariance matrices as

$$C_{\mathrm{L}} = \frac{1}{n_{\mathrm{L}} - 1} \left( B_{\mathrm{L}}^T B_{\mathrm{L}} - \frac{1}{n_{\mathrm{L}}} \left( 1^T B_{\mathrm{L}} \right)^T \left( 1^T B_{\mathrm{L}} \right) \right) \tag{8}$$

$$C_{\mathrm{U}} = \frac{1}{n_{\mathrm{U}} - 1} \left( B_{\mathrm{U}}^T B_{\mathrm{U}} - \frac{1}{n_{\mathrm{U}}} \left( 1^T B_{\mathrm{U}} \right)^T \left( 1^T B_{\mathrm{U}} \right) \right). \tag{9}$$

We then compute the CORAL penalty as

$$\xi\left( B_{\mathrm{L}}, B_{\mathrm{U}} \right) = ||\mu_{\mathrm{L}} - \mu_{\mathrm{U}}||^2 + ||C_{\mathrm{L}} - C_{\mathrm{U}}||_F^2. \tag{10}$$

We adapted our implementation from DomainBed (Gulrajani & Lopez-Paz, 2020), as done in WILDS 1.0. We note that these implementations compute the penalty as a sum of deviations in means and covariances, whereas Sun et al. (2016); Sun & Saenko (2016) penalize deviations in covariances only. (Sun et al. (2016) considers features that are normalized to zero mean.) On applicable datasets, we also strongly augmented all labeled and unlabeled examples using $A_{\mathrm{strong}}$, whereas Sun et al. (2016); Sun & Saenko (2016) do not explicitly require data augmentations. We add augmentations to allow for a fairer comparison to other methods which use augmentations.

Note that CORAL has also been adapted by Gulrajani & Lopez-Paz (2020); Koh et al. (2021) for domain generalization. In particular, where the original CORAL paper defines $L_{\mathrm{penalty}}$ as the divergence between just two kinds of batches (labeled and unlabeled), these works define $L_{\mathrm{penalty}}$ as the divergence between many kinds of batches, where batches are grouped based on domain annotation $d^{(i)}$. For simplicity, we followed the original CORAL formulation and differentiate only between labeled and unlabeled batches. We leave leveraging the domain adaptations $d$ to future work.

APPLICABLE DATASETS. We run CORAL on all datasets except GLOBALWHEAT-WILDS and CIVILCOMMENTS-WILDS. We do not evaluate domain invariant methods on CIVILCOMMENTS-WILDS, since the labeled and unlabeled data are drawn from the same distribution. We do not evaluate CORAL on GLOBALWHEAT-WILDS because CORAL does not port straightforwardly to detection settings.

**DANN.** Algorithm 2 describes DANN, proposed by Ganin et al. (2016). DANN measures the divergence $\xi$ between batches of feature representations using the performance of a discriminator network $h_d$ that aims to discriminate between domains. Given a batch of features (either $B_{\mathrm{L}}$ or $B_{\mathrm{U}}$),

---

**Algorithm 1:** CORAL

---

**Input:** Labeled batch $\{(x_{\mathrm{L}}^{(i)}, y_{\mathrm{L}}^{(i)}, d_{\mathrm{L}}^{(i)}) : i \in (1, \cdots, n_{\mathrm{L}})\}$, unlabeled batch
$\{(x_{\mathrm{U}}^{(i)}, d_{\mathrm{U}}^{(i)}) : i \in (1, \cdots, n_{\mathrm{U}})\}$, strong augmentation function $A_{\mathrm{strong}}$, penalty weight
$\lambda \in \mathbb{R}$, dimension of feature representations $m$

1 Compute feature representations for labeled and unlabeled batches

$$B_{\mathrm{L}} = \{f \circ A_{\mathrm{strong}}(x_{\mathrm{L}}^{(i)}) : i \in (1, \cdots, n_{\mathrm{L}})\}$$
$$B_{\mathrm{U}} = \{f \circ A_{\mathrm{strong}}(x_{\mathrm{U}}^{(i)}) : i \in (1, \cdots, n_{\mathrm{U}})\}$$

2 Compute feature mean and covariances for labeled and unlabeled batches

$$\mu_{\mathrm{L}} = \frac{1}{n_{\mathrm{L}}} 1^T B_{\mathrm{L}}$$
$$\mu_{\mathrm{U}} = \frac{1}{n_{\mathrm{U}}} 1^T B_{\mathrm{U}}$$
$$C_{\mathrm{L}} = \frac{1}{n_{\mathrm{L}} - 1} \left( B_{\mathrm{L}}^T B_{\mathrm{L}} - \frac{1}{n_{\mathrm{L}}} \left(1^T B_{\mathrm{L}}\right)^T \left(1^T B_{\mathrm{L}}\right) \right)$$
$$C_{\mathrm{U}} = \frac{1}{n_{\mathrm{U}} - 1} \left( B_{\mathrm{U}}^T B_{\mathrm{U}} - \frac{1}{n_{\mathrm{U}}} \left(1^T B_{\mathrm{U}}\right)^T \left(1^T B_{\mathrm{U}}\right) \right)$$

3 Update model $h = g \circ f$ on loss

$$\frac{1}{n_{\mathrm{L}}} \sum_{i=1}^{n_{\mathrm{L}}} \ell\left(h \circ A_{\mathrm{strong}}(x_{\mathrm{L}}^{(i)}), y_{\mathrm{L}}^{(i)}\right) + \lambda \left(||\mu_{\mathrm{L}} - \mu_{\mathrm{U}}||^2 + ||C_{\mathrm{L}} - C_{\mathrm{U}}||_F^2\right)$$

---

this deep network $h_d$ must classify whether examples are from the labeled data or unlabeled data. $h_d$ is optimized using a binary classification loss

$$L(h_d) = \frac{1}{n_{\mathrm{L}}} \sum_{i=1}^{n_{\mathrm{L}}} \ell(h_d \circ f \circ A_{\mathrm{strong}}(x_{\mathrm{L}}^{(i)}), 1) + \frac{1}{n_{\mathrm{U}}} \sum_{i=1}^{n_{\mathrm{U}}} \ell(h_d \circ f \circ A_{\mathrm{strong}}(x_{\mathrm{U}}^{(i)}), 0) \qquad (11)$$

The loss of $h_d$ is exactly related to $\xi$ as

$$\xi(B_{\mathrm{L}}, B_{\mathrm{U}}) = -L(h_d) \qquad (12)$$

In other words, at the same time that $h_d$ is optimized to minimize its loss $L(h_d)$, the featurizer $f$ is incentivized to minimize $L_{\mathrm{penalty}} = \xi(B_{\mathrm{L}}, B_{\mathrm{U}}) = -L(h_d)$, or maximize $L(h_d)$. See Algorithm 2 for details.

We adapted our implementation from the Transfer Learning Library (Jiang et al., 2020) and matched all details to the formulation given by Ganin et al. (2016), except for two changes. On applicable datasets, we have strongly all labeled and unlabeled examples using $A_{\mathrm{strong}}$, whereas Ganin et al. (2016) do not explicitly require data augmentations. We add augmentations to allow for a fairer comparison to other methods which use augmentations. Second, where Ganin et al. (2016) optimize $f$, $g$, and $h_d$ using the same learning rate $\eta$, we use three different learning rates $\eta_f, \eta_g, \eta_{h_d}$, following the implementation of the Transfer Learning Library (Jiang et al., 2020).

APPLICABLE DATASETS. We run DANN on all datasets except GLOBALWHEAT-WILDS and CIVILCOMMENTS-WILDS. We do not evaluate domain invariant methods on CIVILCOMMENTS-WILDS, since the labeled and unlabeled data are drawn from the same distribution. We do not evaluate DANN on GLOBALWHEAT-WILDS because DANN does not port straightforwardly to detection settings.

### B.3 SELF-TRAINING METHODS

Self-training methods leverage unlabeled data by "pseudo-labeling" unlabeled examples with the model's own predictions and training on them as if they were labeled examples. In certain for-

---

**Algorithm 2: DANN**

---

**Input:** Labeled batch $\{(x_{\mathrm{L}}^{(i)}, y_{\mathrm{L}}^{(i)}, d_{\mathrm{L}}^{(i)}) : i \in (1, \cdots, n_{\mathrm{L}})\}$, unlabeled batch
$\{(x_{\mathrm{U}}^{(i)}, d_{\mathrm{U}}^{(i)}) : i \in (1, \cdots, n_{\mathrm{U}})\}$, strong augmentation function $A_{\mathrm{strong}}$, penalty weight
$\lambda \in \mathbb{R}$, learning rates $\eta_f, \eta_g, \eta_{h_d}$

1 Compute loss for domain discriminator $h_d$

$$L(h_d) = \frac{1}{n_{\mathrm{L}}} \sum_{i=1}^{n_{\mathrm{L}}} \ell(h_d \circ f \circ A_{\mathrm{strong}}(x_{\mathrm{L}}^{(i)}), 1) + \frac{1}{n_{\mathrm{U}}} \sum_{i=1}^{n_{\mathrm{U}}} \ell(h_d \circ f \circ A_{\mathrm{strong}}(x_{\mathrm{U}}^{(i)}), 0)$$

2 Compute loss for model $h = g \circ f$

$$\frac{1}{n_{\mathrm{L}}} \sum_{i=1}^{n_{\mathrm{L}}} \ell\big(h \circ A_{\mathrm{strong}}(x_{\mathrm{L}}^{(i)}), y_{\mathrm{L}}^{(i)}\big) - \lambda L(h_d)$$

3 Update $f, g, h_d$ using appropriate learning rates $\eta_f, \eta_g, \eta_{h_d}$

---

mulations, this is equivalent to minimizing the model's conditional entropy on the unlabeled data (Grandvalet & Bengio, 2005). Contemporary self-training methods also often make use of consistency regularization, i.e., encouraging the model to make similar predictions on noisy/augmented versions of unlabeled examples. Self-training methods have recently been shown to be empirically successful at unsupervised domain adaptation (Saito et al., 2017; Berthelot et al., 2021; Zhang et al., 2021).

The self-training methods we study follow this general structure: given an unlabeled example $x_{\mathrm{U}}$, these algorithms generate a pseudolabel $\tilde{y}_{\mathrm{U}} = \psi(x_{\mathrm{U}})$, where the pseudolabel-generating function $\psi : \mathcal{X} \to \mathcal{Y}$ differs between algorithms. For classification problems, we study algorithms that produce hard pseudolabels, which are one-hot class predictions, rather than soft pseudolabels, which are continuous distributions over the classes. Next, algorithms define an unlabeled loss $L_{\mathrm{U}}(h)$ for model $h$ by minimizing the loss between pseudolabels $\tilde{y}_{\mathrm{U}}$ and the model's predictions. The algorithms we consider below augment $x_{\mathrm{U}}$ during training; i.e., rather than minimizing the loss between $\tilde{y}_{\mathrm{U}}$ and the model's prediction on $x_{\mathrm{U}}$, the algorithms below minimize the loss of predictions on $A(x_{\mathrm{U}})$, where $A$ is a stochastic, label-preserving augmentation. Assuming model $h$ that outputs real-valued logits, the complete unlabeled loss is

$$L_{\mathrm{U}}(h) = \frac{1}{n_{\mathrm{U}}} \sum_{i=1}^{n_{\mathrm{U}}} \ell\big(h \circ A(x_{\mathrm{U}}^{(i)}), \tilde{y}_{\mathrm{U}}^{(i)}\big) \tag{13}$$

This unlabeled loss is jointly optimized with the standard ERM labeled loss. The balance between the two losses is controlled by hyperparameter $\lambda(t)$, which is a function of the current step $t$.

$$L(h) = L_{\mathrm{L}}(h) + \lambda(t)L_{\mathrm{U}}(h) \tag{14}$$

**Pseudo-Label.** Algorithm 3 describes Pseudo-Label, proposed by Lee (2013). In this algorithm, the model dynamically generates pseudolabels and updates each batch. Formally, the pseudolabel-generating function $\psi$ is given by

$$\tilde{y}_{\mathrm{U}} = \psi(x_{\mathrm{U}}) = \arg\max_y h \circ A_{\mathrm{strong}}(x_{\mathrm{U}})[y] \tag{15}$$

where $A_{\mathrm{strong}}$ is the strong augmentation function described in Appendix C. Pseudo-Label then computes the loss between a strongly augmented example and its associated pseudolabel.

In order to more fairly compare Pseudo-Label to FixMatch, we add on confidence thresholding to the Pseudo-Label algorithm, a feature also added in the implementation of Pseudo-Label by Sohn et al. (2020). When confidence thresholding, examples on which the model has low confidence have zero loss, i.e., for some threshold hyperparameter $\tau$, the loss an example $x_{\mathrm{U}}$ contributes is

$$\mathbf{1}\Big\{\mathrm{Softmax}\Big(\max_y h \circ A_{\mathrm{strong}}(x_{\mathrm{U}})[y]\Big) \geq \tau\Big\} \cdot \ell\big(h \circ A_{\mathrm{strong}}(x_{\mathrm{U}}), y_{\mathrm{U}}\big) \tag{16}$$

Finally, Pseudo-Label increases the balance $\lambda(t)$ between labeled and unlabeled losses over time, initially placing $0$ weight on $L_{\mathrm{U}}(h)$ and then linearly stepping the unlabeled loss weight until it reaches the full value of hyperparameter $\lambda$ at some threshold step. We fix the step at which $\lambda(t)$ reaches its maximum value ($\lambda$) to be $40\%$ of the total number of training steps, matching the implementation of Sohn et al. (2020). This scheduling allows the algorithm to initially prioritize the labeled loss, as generated pseudolabels are mostly incorrect while the model has low accuracy. Formally, at step $t$ and given total number of steps $T$,

$$\lambda(t) = \min\{\frac{t}{0.4T}, 1\} \cdot \lambda \qquad (17)$$

We add augmentations to Pseudo-Label in order to allow for a fairer comparison to other methods that use augmentations. On applicable datasets, we have strongly augmented all labeled and unlabeled examples using $A_{\mathrm{strong}}$, whereas Lee (2013) do not use any data augmentations, i.e., all instances of $A_{\mathrm{strong}}$ are replaced with the identity function.

---

**Algorithm 3:** Pseudo-Label

---

**Input:** Labeled batch $\{(x_{\mathrm{L}}^{(i)}, y_{\mathrm{L}}^{(i)}, d_{\mathrm{L}}^{(i)}) : i \in (1, \cdots, n_{\mathrm{L}})\}$, unlabeled batch
   $\{(x_{\mathrm{U}}^{(i)}, d_{\mathrm{U}}^{(i)}) : i \in (1, \cdots, n_{\mathrm{U}})\}$, strong augmentation function $A_{\mathrm{strong}}$, unlabeled loss weight
   for current step $\lambda(t) \in \mathbb{R}$, confidence threshold $\tau \in [0, 1]$

1 Generate pseudolabels $\tilde{y}_{\mathrm{U}} = \arg\max_y h \circ A_{\mathrm{strong}}(x_{\mathrm{U}})[y]$ for the unlabeled data
2 Update model $h$ on loss

$$\frac{1}{n_{\mathrm{L}}} \sum_{i=1}^{n_{\mathrm{L}}} \ell\big(h \circ A_{\mathrm{strong}}(x_{\mathrm{L}}^{(i)}), y_{\mathrm{L}}^{(i)}\big)$$

$$+ \frac{\lambda(t)}{n_{\mathrm{U}}} \sum_{i=1}^{n_{\mathrm{U}}} \mathbf{1}\Big\{ \mathrm{Softmax}\Big( \max_y h \circ A_{\mathrm{strong}}(x_{\mathrm{U}})[y] \Big) \geq \tau \Big\} \cdot \ell\big(h \circ A_{\mathrm{strong}}(x_{\mathrm{U}}), \tilde{y}_{\mathrm{U}}\big)$$

---

APPLICABLE DATASETS. We evaluate Pseudo-Label on all datasets except POVERTYMAP-WILDS, as POVERTYMAP-WILDS is a regression dataset, and hard pseudolabeling does not port straightforwardly to regression tasks.

**FixMatch.** Algorithm 4 describes FixMatch, proposed by Sohn et al. (2020). Like Pseudo-Label, this algorithm dynamically generates pseudolabels and updates each batch. FixMatch additionally employs consistency regularization on the unlabeled data. While pseudolabels are generated on a weakly augmented view of the unlabeled examples, the loss is computed with respect to predictions on a strongly augmented view. This encourages models to make predictions on a strongly augmented example consistent with its prediction on the same example when weakly augmented. For details about the strong versus weak augmentations we use, see Appendix C.

Formally, the pseudolabel-generating function $\psi$ is given by

$$\tilde{y}_{\mathrm{U}} = \psi(x_{\mathrm{U}}) = \arg\max_y h \circ A_{\mathrm{weak}}(x_{\mathrm{U}})[y] \qquad (18)$$

Like Pseudo-Label, FixMatch uses confidence thresholding, and unlabeled examples on which the model has low confidence have zero loss. Following Sohn et al. (2020), we keep the balance between labeled and unlabeled losses constant at $\lambda(t) = \lambda$. FixMatch's original authors justify keeping $\lambda(t)$ at a fixed magnitude (as opposed to slowly increasing $\lambda(t)$ as in Pseudo-Label) by noting that most predictions made by FixMatch are initially low confidence, so for sufficiently high confidence threshold $\tau$, most unlabeled examples have loss zero, keeping the magnitude of $L_{\mathrm{U}}(h)$ initially small. This magnitude grows over time, providing a natural curriculum (Sohn et al., 2020).

We endeavored to match our implementation of FixMatch to the formulation of Sohn et al. (2020), except in the use of augmentations for labeled data. Where we have strongly augmented all labeled examples using $A_{\mathrm{strong}}$ in Algorithm 4, Sohn et al. (2020) explicitly choose to use weak instead of

strong augmentations on the labeled examples. However, our results on DomainNet in Appendix E suggest that using strong instead of weak augmentations for the labeled examples improves performance, so we use strong augmentations on the labeled examples in order to allow for a fairer comparison to other methods.

---

**Algorithm 4:** FixMatch

---

**Input:** Labeled batch $\{(x_{\mathrm{L}}^{(i)}, y_{\mathrm{L}}^{(i)}, d_{\mathrm{L}}^{(i)}) : i \in (1, \cdots, n_{\mathrm{L}})\}$, unlabeled batch
$\{(x_{\mathrm{U}}^{(i)}, d_{\mathrm{U}}^{(i)}) : i \in (1, \cdots, n_{\mathrm{U}})\}$, weak augmentation function $A_{\mathrm{weak}}$, strong augmentation function $A_{\mathrm{strong}}$, unlabeled loss weight $\lambda \in \mathbb{R}$, confidence threshold $\tau \in [0, 1]$

1 Generate pseudolabels $\tilde{y}_{\mathrm{U}} = \arg\max_y h \circ A_{\mathrm{weak}}(x_{\mathrm{U}})[y]$ for the unlabeled data

2 Update model $h$ on loss

$$\frac{1}{n_{\mathrm{L}}} \sum_{i=1}^{n_{\mathrm{L}}} \ell\big(h \circ A_{\mathrm{strong}}(x_{\mathrm{L}}^{(i)}), y_{\mathrm{L}}^{(i)}\big)$$

$$+ \frac{\lambda}{n_{\mathrm{U}}} \sum_{i=1}^{n_{\mathrm{U}}} \mathbf{1}\Big\{ \mathrm{Softmax}\Big( \max_y h \circ A_{\mathrm{strong}}(x_{\mathrm{U}})[y] \Big) \geq \tau \Big\} \cdot \ell\big(h \circ A_{\mathrm{strong}}(x_{\mathrm{U}}), \tilde{y}_{\mathrm{U}}\big)$$

---

APPLICABLE DATASETS. We evaluate FixMatch on image classification datasets IWILDCAM2020-WILDS, CAMELYON17-WILDS, POVERTYMAP-WILDS, and FMOW-WILDS. We do not evaluate FixMatch on other datasets because FixMatch relies on enforcing consistency across data augmentations, which we only define for image datasets (see Appendix C).

**Noisy Student.** Algorithm 5 describes Noisy Student, proposed by Xie et al. (2020). Unlike Pseudo-Label and FixMatch, which update the model and re-generate new pseudolabels each batch, Noisy Student generates pseudolabels, fixes them, and then trains the model until convergence before generating new pseudolabels. First, an initial teacher model is trained on the labeled data; next, the teacher model pseudolabels the unlabeled data, and a student model is trained on the labeled and pseudolabeled data; finally, the student model becomes the new teacher, and the cycle repeats (see Algorithm 5). Each (teacher, student) pair is termed an *iteration*; we study the results of two iterations.

We train Noisy Student using hard pseudolabels, which the teacher generates over weakly augmented inputs:

$$\tilde{y}_{\mathrm{U}} = \psi(x_{\mathrm{U}}) = \arg\max_y f_{\mathrm{teacher}} \circ A_{\mathrm{weak}}(x_{\mathrm{U}})[y] \qquad (19)$$

While the teacher generates pseudolabels on a weakly augmented data, the student must make both labeled and unlabeled predictions on noisy (i.e., strongly augmented) data. Following Xie et al. (2020), we add a dropout layer ($p = 0.5$) before the student's last layer, randomly corrupting final feature maps. Students thus are trained to be consistent across both data-based and model-based noise. We denote the model with inserted dropout as Dropout $\circ f$. Xie et al. (2020) add even more model-based noise using stochastic depth; for simplicity, we do not use stochastic depth in our implementation.

We follow the original paper and fix the balance between labeled and unlabeled losses as $\lambda(t) = 1$. Noisy Student does not use confidence thresholding.

Note that Xie et al. (2020) use both dropout and strong data augmentations when training the initial teacher on labeled data. We reuse models from our ERM + Data Augmentation experiments as initial teacher models; thus we differ from Xie et al. (2020) in that our initial teachers were trained with strong augmentations, but not dropout (see Algorithm 5).

APPLICABLE DATASETS. We evaluate Noisy Student on all datasets except text datasets CIVILCOMMENTS-WILDS and AMAZON-WILDS. For GLOBALWHEAT-WILDS and OGB-MOLPCBA, we run Noisy Student without noise from data augmentations.

---

**Algorithm 5:** Noisy Student

---

**Input:** Labeled dataset $\{(x_\text{L}, y_\text{L}, d_\text{L})\}$ divided into batches of size $n_\text{L}$, unlabeled dataset $\{(x_\text{U}, d_\text{U})\}$ divided into batches of size $n_\text{U}$, total number of iterations $S$, weak augmentation function $A_\text{weak}$, strong augmentation function $A_\text{strong}$

1 Train an initial teacher model $f^{[0]}$ to convergence on labeled examples using the following batch-wise objective

$$\frac{1}{n_\text{L}} \sum_{i=1}^{n_\text{L}} \ell\big(h \circ A_\text{strong}(x_\text{L}^{(i)}), y_\text{L}^{(i)})\big)$$

**for** *iteration* $s \in (1, \cdots, S)$ **do**

2 Generate fixed pseudolabels $\tilde{y}_\text{U} = \arg\max_y f^{[s-1]} \circ A_\text{weak}(x_\text{U})$ for the unlabeled data

3 Train the next student model $f^{[s]}$ to convergence on unlabeled and labeled examples using the following batch-wise objective

$$\frac{1}{n_\text{L}} \sum_{i=1}^{n_\text{L}} \ell\big(\text{Dropout} \circ h \circ A_\text{strong}(x_\text{L}^{(i)}), y_\text{L}^{(i)})\big) + \frac{1}{n_\text{U}} \sum_{i=1}^{n_\text{U}} \ell\big(\text{Dropout} \circ h \circ A_\text{strong}(x_\text{U}^{(i)}), \tilde{y}_\text{U}^{(i)})\big)$$

---

### B.4 SELF-SUPERVISION METHODS

Self-supervised methods learn useful representations by training on unlabeled data via auxiliary "proxy" tasks. Common approaches include reconstruction tasks (Vincent et al., 2008; Erhan et al., 2010; Devlin et al., 2019; Gidaris et al., 2018; Lewis et al., 2020), which remove or corrupt a small part of each training example and use it as a prediction goal, and contrastive learning (He et al., 2020; Chen et al., 2020b; Caron et al., 2020; Radford et al., 2021b), which aims to learn a representation space such that similar example pairs stay close to each other while dissimilar ones are far apart. The underlying assumption is that feature encoders that solve the proxy tasks will also perform well on the downstream supervised task (Lee et al., 2020a; Wei et al., 2021).

In our work, we consider two self-supervised methods: SwAV Caron et al. (2020) for images and masked language modeling (Devlin et al., 2019) for text. We use these methods to pre-train models on the unlabeled data. In all cases, we start with the same model initialization used for all of the other algorithms on that dataset; do additional pre-training via self-supervision on the unlabeled data; and then initialize a new classifier head and finetune the model via ERM with data augmentation. This follows the procedure in Shen et al. (2021). As a concrete example, for FMOW-WILDS, we use the following procedure to run our ERM experiments:

1. Initialize a DenseNet-121 model (Huang et al., 2017) using ImageNet-pretrained weights.

2. Finetune the model on labeled data from the source domain.

3. Evaluate on held-out data from the target domain.

For SwAV, we use the exact same procedure but with the addition of a second step:

1. Initialize a DenseNet-121 model (Huang et al., 2017) using ImageNet-pretrained weights.

2. Continue pre-training the model with SwAV on unlabeled data from the target domain.

3. Finetune the model on labeled data from the source domain.

4. Evaluate on held-out data from the target domain.

Similarly, for text datasets, we initialized pre-trained BERT models and then continued pre-training them using masked language modeling on the unlabeled data in WILDS 2.0.

We tuned hyperparameters for finetuning, following the exact same procedure and hyperparameters as for ERM, but not for pre-training.

**SwAV.** We directly use the public SwAV repository available at `https://github.com/facebookresearch/swav`. We keep almost all of the hyperparameters used by the original paper for 400 epoch training with batch size 256. However, we make the following changes based on issues and tips from the original authors in the SwAV repository:

1. To stabilize training, we opt not to use a queue; this follows the suggestion in issue #69.

2. For each dataset, we set the number of prototypes to approximately 10x the number of classes; this follows the suggestion in issue #37. For POVERTYMAP-WILDS, which is a regression problem, we use 1000 prototypes, which displayed more stable training than 10 or 100 prototypes.

3. We set $\epsilon = 0.03$ to avoid representation collapse; this follows the suggestion in the Common Issues section of the repository's readme.

4. We set the base learning rate via the suggested "linear scaling" rule (issue #37). In other words, for total batch size (over GPUs) $\geq 512$, the learning rate is scaled linearly. For smaller batch sizes ($< 512$), we set the base learning rate at 0.6. We multiply the base learning rate by $1000\times$ to obtain the final learning rate, since each of the base/final pairs that the paper reports differ by that factor.

We set the maximum number of epochs to 400 but stop pre-training early when the loss does not decrease by more than 0.3% for 5 consecutive epochs.

APPLICABLE DATASETS. We evaluate SwAV on IWILDCAM2020-WILDS, CAMELYON17-WILDS, POVERTYMAP-WILDS, and FMOW-WILDS. We do not evaluate SwAV on other datasets because SwAV relies training with data augmentations, which we only define for image datasets (see Appendix C).

**Masked language modeling (MLM).** MLM is a popular self-supervised objective for text data and is commonly used to pre-train model representations (Devlin et al., 2019). Given an unlabeled text corpus $\mathcal{X} = \{X\}_i$ (e.g., a set of comments for CivilComments; a set of reviews for Amazon), a training example $(x, y)$ can be generated by randomly masking tokens in each text piece $X$ (e.g., $x = $ "The [MASK] is the currency [MASK] the UK"; $y = $ ("pound", "of")). The model is trained to use its representation of the masked input $x$ to predict the original tokens $y$ that should go in each mask. The MLM objective encourages the model to learn syntactic and semantic knowledge (e.g., to predict "of") as well as world knowledge (e.g., to predict "pound") present in the text corpus (Guu et al., 2020).

For our implementation, we use DistilBERT (Sanh et al., 2019) as our initial model and pre-train it with the MLM objective on the unlabeled data of each task (CivilComments, Amazon). Following the original BERT implementation (Devlin et al., 2019), we randomly mask 15% of the tokens in each input text piece, of which 80% are replaced with [MASK], 10% are replaced with a random token (according to the unigram distribution), and 10% are kept unchanged.

APPLICABLE DATASETS. We evaluate masked language modeling on the text datasets CIVILCOMMENTS-WILDS and AMAZON-WILDS.

## C  Data augmentation

In this work, several methods we study leverage data augmentations to encourage generalization across domains. Below, we provide details on our implementations of these augmentations.

**Image classification (iWILDCAM2020-WILDS, CAMELYON17-WILDS, and FMoW-WILDS).** We use a consistent set of data augmentations across image datasets iWILDCAM2020-WILDS, CAMELYON17-WILDS, and FMoW-WILDS. For methods other than SwAV, we define two strengths of data augmentations: a weak function $A_{\text{weak}}$ and a strong function $A_{\text{strong}}$, and we specify both according to Sohn et al. (2020). The weak augmentation function $A_{\text{weak}}$ is a random horizontal flip. The strong augmentation function $A_{\text{strong}}$ is a composition of (i) random horizontal flip, (ii) RandAugment (Cubuk et al., 2020), and (iii) Cutout (DeVries & Taylor, 2017). For the exact implementation of RandAugment, we directly use the implementation of Zhang et al. (2021), which is based on the implementation used by Sohn et al. (2020). This implementation specifies a pool of operations and sample magnitudes for each operation uniformly across a pre-specified range. The pool of operations includes: autocontrast, brightness, color jitter, contrast, equalize, posterize, rotation, sharpness, horizontal and vertical shearing, solarize, and horizontal and vertical translations. We apply $N = 2$ random operations for all experiments (see Appendix D.4).

The labeled loss for all methods, including finetuning models pre-trained with SwAV, uses this strong augmentation function.

For SwAV pre-training, we use the data augmentation pipeline used in the original paper (Caron et al., 2020), which is almost identical to the strong data augmentation introduced in SimCLR (Chen et al., 2020a) but with different random crop scales to accommodate the several additional lower-resolution crops. For each image, the pipeline is the following sequence of random transformations: resized crop, horizontal flip, color jitter, grayscale, and Gaussian blur.

**POVERTYMAP-WILDS.** As POVERTYMAP-WILDS is a dataset of multispectral images, we define a separate set of data augmentations. For methods other than SwAV, we define two strengths of data augmentations: a weak function $A_{\text{weak}}$ and a strong function $A_{\text{strong}}$. The weak augmentation function $A_{\text{weak}}$ is a random horizontal flip. The strong augmentation function $A_{\text{strong}}$ is a composition of (i) random horizontal flip, (ii) random affine transformation, (iii) color jitter on the RGB channels, and (iv) Cutout on all channels (DeVries & Taylor, 2017).

We use the same augmentations for SwAV pre-training as above for iWILDCAM2020-WILDS, CAMELYON17-WILDS, and FMoW-WILDS, but note that the color jitter module is applied only to the RGB channels.

**Other datasets.** We do not define data augmentations for other datasets, i.e., GLOBALWHEAT-WILDS, OGB-MOLPCBA, CIVILCOMMENTS-WILDS, and AMAZON-WILDS. Although GLOBALWHEAT-WILDS is an image dataset and can be transformed using augmentations defined above, we omit data augmentations for simplicity, because such augmentations would generally require changing $y$ as well as $x$ (e.g., random translations on the input image also require translating the bounding box labels). For OGB-MOLPCBA, we omit augmentations because data augmentations on graphs are not well developed. CIVILCOMMENTS-WILDS and AMAZON-WILDS are text datasets; although data augmentations have been proposed for text datasets, we do not use these augmentations because training with augmentations is not as standard on text datasets as on image datasets. For these datasets, methods are benchmarked without augmentations, i.e. we substitute all occurrences of $A_{\text{weak}}, A_{\text{strong}}$ with the identity.

# D EXPERIMENTAL DETAILS

## D.1 IN-DISTRIBUTION VS. OUT-OF-DISTRIBUTION PERFORMANCE

We report both in-distribution and out-of-distribution performance metrics on all datasets, with the exception of OGB-MOLPCBA, which does not have a separate in-distribution test set. Using the terminology in WILDS (Koh et al., 2021), we consider the train-to-train in-distribution comparison on IWILDCAM2020-WILDS, CAMELYON17-WILDS, FMOW-WILDS, and POVERTYMAP-WILDS, and the average comparison on CIVILCOMMENTS-WILDS and AMAZON-WILDS.

## D.2 MODEL ARCHITECTURES

For all experiments, we use the same models for each dataset as in WILDS 1.0:

- IWILDCAM2020-WILDS: ResNet-50 (He et al., 2016).
- CAMELYON17-WILDS: DenseNet-121 (Huang et al., 2017).
- FMOW-WILDS: DenseNet-121 (Huang et al., 2017).
- POVERTYMAP-WILDS: Multi-spectral ResNet-18 (Yeh et al., 2020).
- GLOBALWHEAT-WILDS: Faster-RCNN (Ren et al., 2015).
- OGB-MOLPCBA: Graph Isomorphism Network (Xu et al., 2018).
- CIVILCOMMENTS-WILDS: DistilBERT (Sanh et al., 2019).
- AMAZON-WILDS: DistilBERT (Sanh et al., 2019).

The models for IWILDCAM2020-WILDS, FMOW-WILDS, and GLOBALWHEAT-WILDS were initialized with weights pre-trained on ImageNet. Note that models for CAMELYON17-WILDS were *not* initialized with ImageNet weights. The DistilBERT models were also initialized with pre-trained weights from the Transformers library.

## D.3 BATCH SIZES AND BATCH NORMALIZATION

For each dataset, we set the total batch size (where a batch contains both labeled and unlabeled data) to the maximum that can fit on 12GB of GPU memory (Table 13). For all the methods that leverage unlabeled data, except the pre-training algorithms, we run with 4 steps of gradient accumulation, resulting in a 4× larger effective batch size. For SwAV pre-training, we run with 4 GPUs in parallel, which achieves a similar effect. For masked LM pre-training, we run with the default setting of 256 steps of gradient accumulation. These larger batch sizes deviate from the defaults used in the WILDS paper (Koh et al., 2021). We use these larger batch sizes because methods that leverage unlabeled data tend to use larger batch sizes (Sohn et al., 2020; Xie et al., 2020; Caron et al., 2020).

| Dataset | WILDS 1.0 batch size | WILDS 2.0 batch size |
|---|---|---|
| CAMELYON17-WILDS | 32 | 168 |
| CIVILCOMMENTS-WILDS | 16 | 48 |
| FMOW-WILDS | 32 | 72 |
| POVERTYMAP-WILDS | 64 | 120 |
| AMAZON-WILDS | 8 | 24 |
| IWILDCAM2020-WILDS | 16 | 24 |
| OGB-MOLPCBA | 32 | 4,096 |
| GLOBALWHEAT-WILDS | 4 | 8 |

Table 13: The batch sizes of each dataset from the original WILDS 1.0 paper and the batch sizes used in WILDS 2.0, which correspond to the maximum that can fit into 12GB of GPU memory.

For models that use batch normalization, the composition of each batch affects the way in which batch normalization is applied. For CORAL, DANN, and Pseudo-Label, we concatenate the labeled and unlabeled data together in each batch, so the labeled and unlabeled data are jointly normalized.

| Dataset \ # epochs | ERM | 3:1 ratio | 7:1 ratio | 15:1 ratio |
|---|---|---|---|---|
| IWILDCAM2020-WILDS | 12 | 6 | 3 | 2 |
| CAMELYON17-WILDS | 10 | 5 | 3 | 2 |
| FMOW-WILDS | 60 | 30 | 15 | 8 |
| POVERTYMAP-WILDS | 150 | 75 | 38 | 19 |
| GLOBALWHEAT-WILDS | 12 | 6 | 3 | 2 |
| OGB-MOLPCBA | 200 | 100 | 50 | 25 |
| CIVILCOMMENTS-WILDS | 5 | 3 | 2 | 1 |
| AMAZON-WILDS | 3 | 2 | 1 | 1 |

Table 14: The number of epochs (complete passes over the labeled data) used for each dataset, specified for the ERM baseline as well as different ratios of unlabeled to labeled data within a batch.

For FixMatch, we jointly normalize the labeled data and the strongly augmented unlabeled data, but we separately normalize the weakly augmented unlabeled data in a separate forward pass; we did two forward passes to keep the overall batch sizes consistent with the other algorithms, as in Table 13, while still fitting in GPU memory. For Noisy Student, MLM pre-training, and SwAV pre-training, the unlabeled data is processed separately from the labeled data, so each batch of labeled or unlabeled data is separately normalized.

### D.4 HYPERPARAMETER TUNING

We tune each algorithm separately for each dataset by randomly sampling 10 different hyperparameter configurations within the ranges defined below. We early stop and select the best hyperparameters based on the OOD validation performance, which is computed on the labeled Validation (OOD) data for each dataset; we do not use the labeled Validation (ID) data in our experiments. We then run replicates using the best hyperparameters. For computational reasons, we do not tune hyperparameters for the pre-training algorithms, though we tune the finetuning of their resulting pre-trained models as usual.

**Learning rates.** For all the datasets, except for OGB-MOLPCBA, we multiply the learning rates used in WILDS by the ratio of the effective batch size to the original batch size used in WILDS 1.0. We center the learning rate grid around this modified learning rate $r$, and search over $r \cdot 10^{U(-1,1)}$, where $U$ is the uniform distribution. For OGB-MOLPCBA, we pick $r$ by multiplying the original learning rate by a factor of 10 instead of $4096/32 = 128$ (for ERM, which does not have grandient accumulation), because we found that the latter led to unstable optimization.

**$L_2$-regularization.** Across all datasets and methods, we used the same $L_2$-regularization strengths used in WILDS 1.0.

**Ratio of unlabeled to labeled data in a batch.** For all the domain-invariant and self-training methods, we search over the ratio of unlabeled to labeled data in a batch, using the values $\{3 : 1, 7 : 1, 15 : 1\}$.

**Number of epochs.** We defined an epoch as a complete pass over the labeled data. This means that the number of batches / gradient steps taken per epoch varies with the ratio of unlabeled to labeled data in a batch, as a higher ratio means that each batch contains fewer labeled examples. We adjusted the number of epochs accordingly so that the total amount of compute was similar regardless of the ratio of unlabeled to labeled data in a batch. We allocated roughly twice as much compute (i.e., processing twice as many batches) to methods that used unlabeled data, compared to the purely-supervised ERM baseline. Overall, we set the number of epochs based on the WILDS 1.0 defaults, with some upwards adjustments (due to the different batch sizes and the use of unlabeled data) if we found that the best hyperparameter configuration had not converged on the validation set. Table 14 shows the total number of epochs used per dataset.

## D.5 Algorithm-specific hyperparameters

We tuned the following algorithm-specific hyperparameters:

**CORAL.** We searched over penalty weights $10^{U(-1,1)}$.

**DANN.** We searched over penalty weights $10^{U(-1,1)}$ and have separate learning rates for the featurizer, classifier and domain discriminator. We tuned the learning rate for the classifier and domain discriminator, then fixed the learning rate of the featurizer to be a tenth of the learning rate of the classifier.

**Pseudo-Label, FixMatch, and Noisy Student.** We fixed the penalty weight to be 1. For FixMatch and Pseudo-Label, we searched over confidence thresholds $U(0.7, 0.95)$. Noisy Student does not use a confidence threshold.

**SwAV.** We did not tune SwAV hyperparameters. See Appendix B.4 for a description of the default hyperparameters used.

**Masked language modeling.** We did not tune masked LM hyperparameters, opting instead to use default hyperparameters. For both CIVILCOMMENTS-WILDS and AMAZON-WILDS, we pre-trained DistilBERT for 1,000 steps with a learning rate of $10^{-4}$ and a batch size of 8,192 sequences using gradient accumulation. Following WILDS defaults, for CivilComments, we set the max sequence length to be 300 and for Amazon, 512. We used FP16 training to speed up pretraining.

## D.6 Compute infrastructure

We ran experiments on a mix of NVIDIA GPUs: V100, K80, GeForce RTX, Titan RTX, Titan Xp, and Titan V. SwAV pre-training took approximately 3 days $\times$ 4 V100 GPUs for each dataset, while masked LM pre-training took approximately 3 days for a single GPU for each dataset. The other algorithms took less than a day on a V100 to run. The runtime estimates in Section 6 are estimated for V100 GPUs. We used the Weights and Biases platform (Biewald, 2020) to monitor experiments.

# E    EXPERIMENTS ON DOMAINNET

Prior work has shown that domain-invariant, self-training, and self-supervised methods can perform well on standard benchmarks for unsupervised domain adaptation. In this section, we describe our experiments on DomainNet (Peng et al., 2019), a standard unsupervised domain adaptation benchmark for object recognition. Our goal was to verify that our training/tuning protocol and our implementations of the methods we benchmark in Section 6—which differ slightly from prior work in the ways described in Section B—still result in models that can perform well on DomainNet. Consistent with prior work, the methods we benchmark in Section 6, with the exception of CORAL, all improve over standard ERM training in our DomainNet experiments.

## E.1    SETUP

DomainNet is an object recognition dataset with approximately 600,000 images across six different domains: *sketch*, *real*, *quickdraw*, *painting*, *infograph* and *clipart* (Peng et al., 2019). Typically, one of these domains is selected as the source, and another domain as the target for evaluation. In our experiments, we use the *real → sketch* setting for two reasons: it is a common choice in prior work on DomainNet, and as our models are pre-trained on ImageNet (following (Zhang et al., 2021)), we wanted to use the *real* domain as the source to be consistent with the realistic photographs used for ImageNet pretraining. While it is common to evaluate methods on multiple pairs of source and target domains in DomainNet, in our experiments we only chose one pair, as our goal was only to verify consistency with prior results.

**Data.**    The DomainNet dataset includes train and test splits for each of the domains, with 70%–30% split between train and test examples. The *real* domain has 172,947 images total: 120,906 images in the train split and 52,041 images in the test split. The target domain, *sketch*, has a total of 69,128 images: 48,212 in the train split and 20,916 images are in the test split. We used this data in the following way:

1. For **training**, we used the source training examples (with labels) and the target training examples (without labels).

2. For **validation**, we used the same set of target training examples, but with labels; this overestimates performance in a true domain adaptation setting (where one would not have labeled target data), but it is a common practice in the literature, and we followed it for consistency with Jiang et al. (2020) and Zhang et al. (2021).

3. For **evaluation**, we used the source test examples as the in-distribution test set, and the target test examples as the out-of-distribution test set.

**Hyperparameters and other details.**    Other experimental details followed our main experiments in Section 6. We used the strong and weak data augmentation described for image classification in Appendix C. We set the total batch size to 96, which is the maximum that can fit on 12GB of GPU memory. We tuned hyperparameters with the protocol described in Section D.4. Specifically, for all methods, we fixed $L_2$-regularization at $10^{-4}$. We then randomly sampled learning rates from $10^{U(-4,-2)}$ to train the ERM with data augmentation model. For all other models, we took the best learning rate that we found for the ERM with data augmentation model and searched over one order of magnitude lower and higher from it. As in Zhang et al. (2021), we used a ResNet-50 model initialized by pretraining on ImageNet. For SwAV pre-training, instead of following the early-stopping procedure in Appendix B.4, we trained for the full 400 epochs used in Caron et al. (2020) since the experiment finished relatively quickly compared to the larger WILDS 2.0 datasets.

## E.2    RESULTS

Table 15 shows the results of our experiments on *real → sketch*. The use of (strong) data augmentation improved ERM performance from 34.9% to 35.9%. All unsupervised adaptation methods except CORAL improved over ERM. We also tested the use of strong vs. weak augmentation for labeled examples for both Pseudo-Label and FixMatch, and we found that using strong augmentation for the labeled examples improves performance.

|  | In-distribution (real) | Out-of-distribution (sketch) |
|---|---|---|
| ERM (-data aug) | 82.6 (0.0) | 34.9 (0.2) |
| ERM | 82.5 (0.3) | 35.9 (0.3) |
| CORAL | 79.1 (0.4) | 33.6 (0.6) |
| DANN | 77.8 (0.2) | 39.4 (0.8) |
| Pseudo-Label | 79.9 (0.2) | 36.1 (0.4) |
| Pseudo-Label (weak aug) | 79.9 (0.6) | 32.0 (0.8) |
| FixMatch | 80.8 (0.2) | **50.2** (0.4) |
| FixMatch (weak aug) | 80.1 (0.1) | 49.3 (0.2) |
| Noisy Student | 82.0 (0.3) | 39.7 (0.2) |
| SwAV | 79.0 (0.3) | 38.2 (0.4) |

Table 15: The in-distribution vs. out-of-distribution test performance of each method on DomainNet (*real* → *sketch*). We also included the results of applying weak instead of strong augmentation on labeled examples for Pseudo-Label and FixMatch. Parentheses show standard deviation across 3 replicates.

For DANN, Pseudo-Label, and FixMatch, we compared our results against the results reported in Zhang et al. (2021). Performance was similar for DANN (ours, 39.4%; theirs, 40.0%). For Fix-Match, our implementation performs better (ours, 50.2%; theirs, 45.3%); this is partially due to our use of strong instead of weak augmentation for the labeled data, which increases performance by 0.9%. For Pseudo-Label, our implementation performs worse (ours, 36.1%; theirs, 40.6%), and we believe it is due to variation in hyperparameter tuning.

For Noisy Student, Berthelot et al. (2021) reported significantly lower numbers (ours, 39.7%; theirs, 32.6%). However, this is expected as they trained their models from scratch, whereas we used ImageNet-pretrained models.

We were unable to find comparable results in prior work for CORAL and SwAV pretraining on the *real* → *sketch* split. Prior work has shown that these methods can improve performance on other unsupervised adaptation datasets (Sun & Saenko, 2016; Shen et al., 2021). On our DomainNet experiments, we found that SwAV pre-training did improve performance over ERM, though CORAL did not (Table 15).

## F Fully-labeled ERM experimental details

The self-training methods we evaluate in Section 5 generate a pseudolabel $\tilde{y}_U$ for each unlabeled example $x_U$ and then train on $(x_U, \tilde{y}_U)$ as if the pseudolabels were true labels. However, these pseudolabels may not be accurate. In this section, we describe how we ran fully-labeled ERM experiments using ground truth labels on the "unlabeled" data to establish informal upper bounds on how well we might expect a standard self-training approach to perform with perfect pseudolabel accuracy.

For four of our datasets (Amazon-wilds, CivilComments-wilds, iWildCam2020-wilds, and FMoW-wilds), we curated the "unlabeled" data by taking labeled examples and discarding the ground truth labels. For example, all 268,761 of the unlabeled target reviews in Amazon-wilds actually have associated star ratings; these are available in our data loaders, but in our main experiments we treat these reviews as unlabeled by not loading the star ratings. We evaluated models trained via empirical risk minimization (ERM) on the combination of the standard labeled training set and the unlabeled data with these hidden labels revealed. For example, in Amazon-wilds, we pool together the labeled source examples as well as the unlabeled target examples with ground truth labels, and evaluate ERM models trained on all of that data. As with all of the other experiments in this paper, we evaluate test performance for all datasets on the labeled target splits, so at no point are we training on our actual test examples.

### F.1 Hyperparameters

**Pooling labeled and unlabeled data.** For all datasets, we pooled labeled source examples with examples from the same "unlabeled" split as in our main experiments (Table 2). We computed gradients for labeled minibatches and unlabeled minibatches separately, which means that for models using batch normalization, the labeled and unlabeled data were normalized separately. However, we fixed the labeled to "unlabeled" batch size ratios to match the overall labeled to unlabeled dataset size ratio, so other than the batch normalization effects, the training procedure can be viewed as running ERM on the pooled labeled and "unlabeled" data.

**Number of epochs.** With the exception of iWildCam2020-wilds, detailed below, we followed the procedure in Appendix D.4 to adjust the number of epochs based on the labeled to unlabeled batch size ratios. This resulted in a similar amount of computation allocated to these fully-labeled ERM experiments as the other experiments in Table 2.

**Other details.** Other experimental details were kept similar to the other experiments in the paper. Specifically, we tuned each experiment by randomly sampling 10 different hyperparameters within the ranges defined in Appendix D.4; the only hyperparameter we tuned in these experiments was the learning rate. We early stopped and selected the best hyperparameters based on the OOD validation performance, and then ran replicates using the best hyperparameters. We also used data augmentation for iWildCam2020-wilds and FMoW-wilds but not for Amazon-wilds and CivilComments-wilds.

### F.2 Dataset-specific details

**Amazon-wilds.** We matched the experiments in Table 2 by training on the unlabeled target data (268,761 examples). In addition, we ran a separate experiment where we trained on the unlabeled extra data instead of the unlabeled target data, as the former has $10\times$ the number of examples (2,927,841 examples). However, this did not improve performance. Using the unlabeled target data, we obtained an average accuracy of 73.6 ($\pm$ 0.1) and a 10th percentile accuracy of 56.4 ($\pm$ 0.8), whereas using the unlabeled extra data, we obtained an average accuracy of 73.1 ($\pm$ 0.1) and a 10th percentile accuracy of 54.7 ($\pm$ 0.0).

**CivilComments-wilds.** We used the unlabeled extra split (1,551,515 examples). As in our other experiments on CivilComments-wilds, we accounted for label imbalance by sampling class-balanced labeled and "unlabeled" batches during training.

**IWILDCAM2020-WILDS.** We used the unlabeled extra split. Out of the 819,120 unlabeled extra examples, 108,452 examples have ground truth labels (animal species) that are not present in the labeled training and test sets, so we omitted those examples and trained on the remaining 710,668 examples. We found that we required twice as many epochs compared to the other unlabeled methods for the fully-labeled ERM training to converge, so we doubled the amount of compute allocated to the fully-labeled IWILDCAM2020-WILDS experiments.

**FMOW-WILDS.** We used the unlabeled target split (173,208 examples).

## G    USING THE WILDS LIBRARY WITH UNLABELED DATA

We have extended the existing WILDS library (Koh et al., 2021) to add data loaders for each of the 8 datasets with unlabeled data. These data loaders are compatible with the WILDS 1.0 APIs, allowing the unlabeled data to be accessed in a similar way to the labeled data:

```python
>>> from wilds import get_dataset
>>> from wilds.common.data_loaders import get_train_loader
>>> import torchvision.transforms as transforms
# Load the labeled data
>>> dataset = get_dataset(dataset="fmow", download=True)
>>> labeled_subset = dataset.get_subset("train", transform=transforms.ToTensor())
>>> data_loader = get_train_loader("standard", labeled_subset, batch_size=16)
# Load the unlabeled data
>>> dataset = get_dataset(dataset="fmow", unlabeled=True, download=True)
>>> unlabeled_subset = dataset.get_subset("test_unlabeled", transform=transforms.ToTensor())
>>> unlabeled_data_loader = get_train_loader("standard", unlabeled_subset, batch_size=64)
# Train loop
>>> for labeled_batch, unlabeled_batch in zip(data_loader, unlabeled_data_loader):
...     x, y, metadata = labeled_batch
...     unlabeled_x, unlabeled_metadata = unlabeled_batch
...     ...
```

Figure 3: Example of data loading for both labeled and unlabeled data.

As in the existing WILDS library, data downloading is automated. In addition, we implemented CORAL, DANN, Pseudo-Label, FixMatch, and Noisy Student using the existing WILDS interfaces. This allows developers to easily extend these algorithms and evaluate them in a standardized way on all of the WILDS datasets with unlabeled data. The WILDS repository also contains scripts for masked language model pre-training and for SwAV pre-training, which uses a modified version of the public SwAV repository that can interface with the WILDS data loaders.

