# OpenReview forum: "Extending the WILDS Benchmark for Unsupervised Adaptation"
_ICLR.cc/2022/Conference — ICLR 2022 Oral_

### Official Review · Reviewer_NvLs · 2021-10-27

**Correctness:** 4
**Technical Novelty And Significance:** 2
**Empirical Novelty And Significance:** 3
**Recommendation:** 8
**Confidence:** 3

**Main Review:**

Strength:
- The work behind the data collection and baseline computation appears sizeable and very valuable
- The authors implemented many of the top-performing algorithms meant to address domain shift with unlabelled data. They provide their implementation, as well as a unified, didactic, and detailed explanation.
- The conclusions are both measured and essential: we lack a definitive answer to domain shift in the wild, and unlabelled only provides a very partial solution.
- The writing is remarkably clear and to the point.

Weakness:
I am struggling to fault the paper.  The extensive appendix answered most of my questions.
- The paper does not have a methodological contribution per se, but the answers provided from the meta-analysis are new - at least to me, even if the fact that current methods would not hold too well for in-the-wild data was suspected
- I have some questions about missing details, detailed below

Questions:
- "Models are trained on labeled data from the source domains, as well as unlabeled data from one or more of the other sources, depending on what is realistic for the application." I couldn't understand which problems were allowed to use OOD unlabelled data for training and which ones were not. From my understanding, unlabelled data from all domains (source, val, target, extra) are merged and used for training; is this correct? And what would be the use of unlabelled data if not for helping the training?
- As a follow-up, are unlabelled data from all domains (ID/OOD) used identically by all methods? DANN, for example, may work better if only using target domain unlabelled data, if available.
- The justification to not use augmentation on GlobalWheat are weak: translating and rotating bounding boxes seems easy enough?
- It seems that you did not try approaches that explicitly use the domain as weak labels or adversarially, such as "Adversarial Multiple Source Domain Adaptation, Zhao et al. NeurIPS2018. I am not suggesting that you do (the comparison is already more than sufficient), but you should mention these approaches. Note that these would only work when the domains are not too numerous and meaningful.

Details:
The domains of PovertyMap are missing in Fig2

**Summary Of The Paper:**

The authors present U-WILDS, an extension of the multi-task, large-scale domain-shift dataset WILDS. They provide a large quantity of unlabelled data complementing 8 of the existing multidomain labeled datasets in WILDS. They propose an extensive array of experiments evaluating the ability of a wide variety of algorithms to leverage the unlabelled data to address domain-shift. They present reasoned conclusions, and open-source datasets and implementations.


**Summary Of The Review:**

An impressive and exemplary dataset paper. The data collection and baseline implementation could be an important stepping stone for the ML community to address its biggest challenge yet: domain shift in the Wild. This paper highlights that we are not there yet, and that unlabelled data give an encouraging venue that has not reached maturity yet.

---

> ### Author Response · Authors · 2021-11-18
> **Response to Reviewer NvLs**
>
> We thank the reviewer for their encouraging feedback and suggestions, and we are glad that the reviewer appreciated our attempt to explain each method and our implementation in a unified manner. Below, we respond to their questions in turn.
>
> ---
>
> > Correctness: 1: The main claims of the paper are incorrect or not at all supported by theory or empirical results.
>
> May we politely ask if this rating was made in error, as the text of the review does not seem to match the rating? If so, would the reviewer mind changing the rating? If not, please let us know which claims are unsupported and we would be glad to clarify or edit them. Thank you!
>
> ---
>
> > "Models are trained on labeled data from the source domains, as well as unlabeled data from one or more of the other sources, depending on what is realistic for the application." I couldn't understand which problems were allowed to use OOD unlabelled data for training and which ones were not. From my understanding, unlabelled data from all domains (source, val, target, extra) are merged and used for training; is this correct? And what would be the use of unlabelled data if not for helping the training?
>
> We apologize for the lack of clarity. Other than the ERM models, all of the other methods used OOD unlabeled data for training. For our experiments, we picked a single type of unlabeled data (source, val, target, extra) for each dataset -- this is marked just below each dataset name in Table 2 -- and did not use the other types of unlabeled data. We did this for simplicity, but our public codebase will allow researchers to use whichever types of unlabeled data they would like, including merging and using all of them, or using them separately to compare the relative benefits of training with one type of unlabeled data versus another. We have clarified this in Section 6.1.
>
> ---
>
> > As a follow-up, are unlabelled data from all domains (ID/OOD) used identically by all methods? DANN, for example, may work better if only using target domain unlabelled data, if available.
>
> Yes, that is right: the unlabeled data is used identically by all methods (except ERM, which does not use any unlabeled data). For example, for iWildCam, all methods use unlabeled data from the extra domains; while for Camelyon, all methods use unlabeled data from the target domains. We agree with the reviewer that different types of unlabeled data (e.g., extra vs. target) might work better for some methods than others. In general, for our experiments, we picked the unlabeled target data whenever available, because we thought that this was likely to lead to stronger results. We have also clarified this in Section 6.1.
>
> ---
>
> > The justification to not use augmentation on GlobalWheat are weak: translating and rotating bounding boxes seems easy enough?
>
> We agree with the reviewer that it is straightforward to apply standard transformations, like translation and rotation, to bounding boxes. Our concern was that doing so would break the consistency assumption made by algorithms like FixMatch and NoisyStudent. For example, those algorithms assume that the predictions made on any example should be invariant under data augmentation, but if the data augmentation also changes the labels (e.g., by translating the image in such a way as to remove a bounding box), then this assumption would no longer be true. It did not seem straightforward to us to modify these algorithms accordingly. Another possibility would have been to define a custom set of augmentations that didn’t change the bounding boxes (e.g., just changing the color of the image, or adding some sort of random noise), but to our knowledge, that is not a common approach taken in detection tasks. To be clear, we believe that it would be an interesting research direction to experiment with methods that can exploit label-changing augmentations, but for the reasons above, we decided that it was out of the scope of the current paper. We have clarified this in the GlobalWheat paragraph in Section 6. Thank you for bringing this up.
>
> ---
>
> > It seems that you did not try approaches that explicitly use the domain as weak labels or adversarially, such as "Adversarial Multiple Source Domain Adaptation, Zhao et al. NeurIPS2018. I am not suggesting that you do (the comparison is already more than sufficient), but you should mention these approaches. Note that these would only work when the domains are not too numerous and meaningful.
>
> We agree with the reviewer that approaches that explicitly use domain annotations are promising, and we thank them for the pointer! We have edited the last paragraph of our discussion (Section 7) to expand on this point accordingly.
>
> ---
>
> > Details: The domains of PovertyMap are missing in Fig2
>
> Thank you! We apologize for the omission and have fixed it in the revision.

---

> > ### Comment · Reviewer_NvLs · 2021-11-20
> > **Correction**
> >
> > I apologize for having misclicked the "correctness" setting. I naturally meant to select the option 4.
> >
> > My main concerns have all been adressed, and the paper is now clearer. I commend the authors for their work in editing the paper and responding to the reviews.

---

> > > ### Author Response · Authors · 2021-11-20
> > > **Thank you**
> > >
> > > Thank you very much. We appreciate it!

---

### Official Review · Reviewer_RpU9 · 2021-11-02

**Correctness:** 3
**Technical Novelty And Significance:** 3
**Empirical Novelty And Significance:** 3
**Recommendation:** 8
**Confidence:** 4

**Main Review:**

### Strengths

- The dataset is a useful addition to the WILDS dataset, and the provided unlabeled data would be very useful to design many unsupervised generalization algorithms.
- The paper is very well written, easy to understand and well organized, although the experimental evaluation could be explained in greater detail in main paper.

### Required Clarifications

- The models chosen for benchmarking the methods are not suitable. While domain adaptation methods like DANN and CORAL work strictly with an assumption of single source to target domains, semi-supervised and self-supervised works are well-known to work only for within-domain samples. In this respect, the observations found with respect to the benchmarking, although useful, aren't surprising.
- self-supervised, semi-supervised and DA methods make different assumptions about the availability of target data. self-supervised requires target labels for fine-tuning, while DA methods only require unlabeled target data during training and nothing more. So, comparing both of them together using a common yard-stick is, in my opinion, not appropriate.
- The main paper needs to have more experimental detail such as what are the exact labeled and unlabeled data from source and target for each of the experiments.
- The DA methods considered are quite old and primitive. The authors are encouraged to consider more recent benchmarks like CDAN, CAN, AFN etc.
- The authors are also encouraged to use multi-source/multi-domain adaptation benchmarks [1,2,3] which seem most appropriate for the dataset proposed. at least for the visual recognition datasets. While self-supervised and semi-supervised methods aren't particularly designed keeping in mind cross-domain transfer, UDA methods like DANN and CORAL aren't designed to handle multi-domains.
- The authors are also encouraged to compare and contrast with [1] in terms of the datasets proposed in both papers (although these can be considered contemporary works).
- The key takeaways from sec 7 are not that surprising, and things like lack of augmentation strategies for many modalities and model selection for DA  are already well known to the community, while ineffectiveness of pre-training seems interesting.

1. Dubey, Abhimanyu, et al. "Adaptive Methods for Real-World Domain Generalization." _Proceedings of the IEEE/CVF Conference on Computer Vision and Pattern Recognition_. 2021.
2. Li, Ya, et al. "Deep domain generalization via conditional invariant adversarial networks." _Proceedings of the European Conference on Computer Vision (ECCV)_. 2018.
3. Li, Da, et al. "Deeper, broader and artier domain generalization." _Proceedings of the IEEE international conference on computer vision_. 2017.

**Summary Of The Paper:**

### New large scale dataset for unsupervised transfer learning

The paper proposed an extension to the popular WILDS benchmark dataset by augmenting the different domain data with additional unlabeled examples. The dataset consists of data of various modalities including images, graph and text from various domains. Additionally, many recent methods that leverage unlabeled images to achieve generalization are bench marked on the proposed datasets.

**Summary Of The Review:**

Although the dataset itself is very useful and important, the analysis and bench marking needs some improvement. Nevertheless, this paper definitely has merit, and if the authors could clarify the questions raised, I would be happy to update the score. I haven't gone through the supplementary material in detail. If any of the questions above have a direct answer in the suppl. material, the authors can directly point to that and I would be happy to update my comments.

Post Rebuttal
****************

I thank the authors for answering all my queries patiently, which answered most of the questions I had regarding suitability of self-supervised and semi-supervised algorithms to the task. I would like to raise my score, and suggest the authors to include these discussions in the main paper.

---

> ### Author Response · Authors · 2021-11-18
> **Response to Reviewer RpU9, part 1/2**
>
> We thank the reviewer for their detailed and thoughtful feedback. Below, we address them in turn.
>
> ---
>
> > The models chosen for benchmarking the methods are not suitable. While domain adaptation methods like DANN and CORAL work strictly with an assumption of single source to target domains, semi-supervised and self-supervised works are well-known to work only for within-domain samples. In this respect, the observations found with respect to the benchmarking, although useful, aren't surprising.
>
> We agree with the reviewer that many of the semi-supervised and self-supervised methods were originally developed for within-domain samples. However, recent work has shown that semi-supervised and self-supervised methods can actually be highly effective on standard domain adaptation benchmarks. For example, Zhang et al. [2] and Berthelot et al. [3] show that FixMatch and NoisyStudent (which we benchmark on U-WILDS) can outperform strong UDA methods like MCD on domain adaptation benchmarks like Digit-Five and DomainNet. As further examples, on the semi-supervised / self-training side:
> - Saito et al. [1] introduce a pseudo-labeling technique that can outperform methods like DANN and MMD on digits datasets.
> - Zhang et al. [2] show that methods such as FixMatch and pseudo-labeling (which are included in the methods we tested for U-WILDS) can outperform UDA methods such as DANN, CDAN, AFN, and MDD on benchmarks like VisDA-2017 and DomainNet.
> - Berthelot et al. [3] show that self-training methods like AdaMatch, FixMatch, and NoisyStudent can substantially outperform UDA methods like MCD on digits datasets as well as DomainNet.
>
> On the self-supervised side, for example:
> - Wang et al. [4] show that self-supervised contrastive learning approaches can outperform UDA approaches on VisDA-2017 and Office-31.
> - Tsai et al. [5] show that self-supervised contrastive learning approaches can help to train models to be robust to spurious domain-specific correlations, enabling them to extrapolate out-of-domain more reliably.
>
>
> In our revision, we also ran additional experiments on the real -> sketch split in DomainNet to confirm that the changes we made to the methods we tested (for standardization and consistency) did not affect their performance. As in prior work, our results show that **all of the methods we benchmark on U-WILDS outperform ERM on this DomainNet split.**
>
> |                 | In-dist (real) | Out-of-dist (sketch) |
> |-----------------|----------------|----------------------|
> | ERM (-data aug) | 82.6 (0.0)     | 34.9 (0.2)           |
> | ERM             | 82.5 (0.3)     | 35.9 (0.3)           |
> | CORAL           | 78.0 (0.5)     | 37.3 (0.4)           |
> | DANN            | 77.8 (0.2)     | 39.4 (0.8)           |
> | Pseudo-Label    | 79.9 (0.2)     | 36.1 (0.4)           |
> | FixMatch        | 80.8 (0.2)     | **50.2 (0.4)**       |
> | Noisy Student   | 82.0 (0.3)     | 39.7 (0.2)           |
> | SwAV            | 79.0 (0.3)     | 38.2 (0.4)           |
>
> **However, the benchmarked methods do not do as well on U-WILDS.** We believe that our observations are interesting in light of the strong prior results on these methods on other benchmarks. For example, FixMatch does well on VisDA-2017, DomainNet, and Digit-Five [2,3], and it does extremely well on the DomainNet experiments we ran above, but its performance is starkly worse on the U-WILDS datasets, where it does not improve performance over ERM.
>
> We have edited Sections 5 and 6 to highlight our reasoning and the prior work cited above, as we agree with the reviewer that it is otherwise unclear. We have also described the DomainNet setup and results in more detail in Appendix F. We thank the reviewer for bringing this point up.
>
> [1] Saito, K., Ushiku, Y., & Harada, T. (2017). Asymmetric tri-training for unsupervised domain adaptation. In International Conference on Machine Learning (pp. 2988-2997). PMLR.
>
> [2] Zhang, Y., Zhang, H., Deng, B., Li, S., Jia, K., & Zhang, L. (2021). Semi-supervised Models are Strong Unsupervised Domain Adaptation Learners. arXiv preprint arXiv:2106.00417.
>
> [3] Berthelot, D., Roelofs, R., Sohn, K., Carlini, N., & Kurakin, A. (2021). AdaMatch: A Unified Approach to Semi-Supervised Learning and Domain Adaptation. arXiv preprint arXiv:2106.04732.
>
> [4] Wang, R., Wu, Z., Weng, Z., Chen, J., Qi, G. J., & Jiang, Y. G. (2021). Cross-domain Contrastive Learning for Unsupervised Domain Adaptation. arXiv preprint arXiv:2106.05528.
>
> [5] Tsai, Y. H. H., Ma, M. Q., Zhao, H., Zhang, K., Morency, L. P., & Salakhutdinov, R. (2021). Conditional Contrastive Learning: Removing Undesirable Information in Self-Supervised Representations. arXiv preprint arXiv:2106.02866.

---

> > ### Author Response · Authors · 2021-11-18
> > **Response to Reviewer RpU9, part 2/2**
> >
> > > Self-supervised, semi-supervised and DA methods make different assumptions about the availability of target data. Self-supervised requires target labels for fine-tuning, while DA methods only require unlabeled target data during training and nothing more. So, comparing both of them together using a common yard-stick is, in my opinion, not appropriate.
> >
> > In our benchmark, all of our methods use the exact same labeled and unlabeled data for each dataset. In particular, none of them use any target labels. We apologize for the lack of clarity, and we have edited the setup in Section 6.1 to clarify this. As we discussed above, prior work has shown that self-supervised and semi-supervised methods can perform well on domain adaptation benchmarks with nothing more than labeled source data and unlabeled target data.
> >
> > ---
> >
> > > The DA methods considered are quite old and primitive. The authors are encouraged to consider more recent benchmarks like CDAN, CAN, AFN etc. The authors are also encouraged to use multi-source/multi-domain adaptation benchmarks [1,2,3] which seem most appropriate for the dataset proposed. at least for the visual recognition datasets. While self-supervised and semi-supervised methods aren't particularly designed keeping in mind cross-domain transfer, UDA methods like DANN and CORAL aren't designed to handle multi-domains.
> >
> > We agree with the reviewer that newer and more sophisticated domain adaptation methods, and in particular those that can exploit multiple domains, might perform better on U-WILDS. In our revision, we have expanded on the discussion of other domain invariance methods in Appendix B.2, including citations to CDAN, CAN, and AFN, and a discussion of multi-source methods. We have also edited the main text in Section 5 to point to this explicitly. We hope that by releasing the U-WILDS open-source package and public leaderboard, we can encourage the broader research community to test out other existing methods and develop new methods on the U-WILDS datasets.
> >
> > At least on benchmarks like DomainNet and VisDA-2017, we note that the prior work cited above [1-5] suggest that the self-training and self-supervised methods we study can outperform existing domain-invariance methods. However, we agree with the reviewer that newer domain-invariance methods would still be interesting to try.
> >
> > ---
> >
> > > The authors are also encouraged to compare and contrast with [1] in terms of the datasets proposed in both papers (although these can be considered contemporary works).
> >
> > We thank the reviewer for the pointer. The Geo-YFCC dataset introduced in that paper seems like a great dataset for domain generalization in the context of object recognition, and we have added it to the related work (Section 2). The U-WILDS datasets differ from Geo-YFCC in two ways. First, the U-WILDS data splits contain both labeled and unlabeled data, which allows them to be used to evaluate domain adaptation algorithms. In contrast, Geo-YFCC is geared for the transductive test-time adaptation setting, where they assume that models do not have access to any unlabeled data at training time, but are allowed to adapt on-the-fly to a small number of unlabeled test samples after training. Second, Geo-YFCC focuses on object recognition with ImageNet classes, which is an important but comparatively well-studied problem. For U-WILDS, we have deliberately aimed to broaden the range of modalities and applications studied.
> >
> > ---
> >
> >  > The key takeaways from sec 7 are not that surprising, and things like lack of augmentation strategies for many modalities and model selection for DA are already well known to the community, while ineffectiveness of pre-training seems interesting.
> >
> > In addition to the ineffectiveness of pre-training, we believe that one of the more surprising results is how methods that can obtain state-of-the-art performance on standard domain adaptation benchmarks (such as FixMatch on DomainNet) can nonetheless fail to improve performance on the U-WILDS datasets, whereas other methods that are conceptually quite similar, like Noisy Student, can help in some settings.
> >
> > ---
> >
> > We appreciate the detailed clarifications requested by the reviewer, and we hope that our answers and revisions have adequately addressed them. Please let us know if there are any other questions or comments that we might be able to get to in the discussion period. Thank you.

---

> > > ### Comment · Reviewer_RpU9 · 2021-11-20
> > > **Response to the rebuttal**
> > >
> > > Thanks for your detailed rebuttal.
> > >
> > > I still fail to understand how can domain adaptation and self-supervised algorithms work with similar assumptions about labeled and unlabeled data? Self-supervised algorithms like SwAV treat a source domain as unlabeled in the first phase of training, but finetune on some amount of labeled data on a downstream task. Domain adaptation treats source domain as labeled and target domain as unlabeled. Perhaps the authors could clarify this a bit more.
> > >
> > > "In particular, none of them use any target labels". Can you explain how is SwAV trained in this setting?
> > >
> > > nevertheless, it is indeed surprising to know that semi-supervised and self-supervised models are quite good OOD generalizers. In that light, I would like to change my statements and agree that the experiment results are indeed significant.  you should definitely add this discussion to the paper.

---

> > > > ### Author Response · Authors · 2021-11-20
> > > > **Clarifying self-supervised algorithms**
> > > >
> > > > Thank you for the quick reply!
> > > > > I still fail to understand how can domain adaptation and self-supervised algorithms work with similar assumptions about labeled and unlabeled data? Self-supervised algorithms like SwAV treat a source domain as unlabeled in the first phase of training, but finetune on some amount of labeled data on a downstream task. Domain adaptation treats source domain as labeled and target domain as unlabeled. Perhaps the authors could clarify this a bit more. "In particular, none of them use any target labels". Can you explain how is SwAV trained in this setting?
> > > >
> > > > We apologize for the lack of clarity about self-supervised algorithms like SwAV. The reviewer is correct that such algorithms are typically pre-trained on unlabeled data and then finetuned on labeled data on a downstream task, which might or might not come from the same data distribution as the unlabeled data. In our case, we are similarly treating self-supervision as a means of pre-training, except that we pre-train on the target unlabeled data, and then finetune on the source labeled data (since that is the only labeled data that we have). As a concrete illustration using the DomainNet real -> sketch experiment, training a standard ERM model would involve the following:
> > > >
> > > > 1. Initialize the model using ImageNet-pretrained weights
> > > > 2. Finetune the model on labeled data from the source (“real”) domain
> > > > 3. Evaluate on held-out data from the target (“sketch”) domain
> > > >
> > > > while using SwAV would involve adding the second step in the following (all other steps are identical):
> > > >
> > > > 1. Initialize the model using ImageNet-pretrained weights
> > > > 2. Continue pre-training the model with SwAV on unlabeled data from the target (“sketch”) domain
> > > > 3. Finetune the model on labeled data from the source (“real”) domain
> > > > 4. Evaluate on held-out data from the target (“sketch”) domain
> > > >
> > > > The DomainNet experiments that we added in the revision (Table 15) show that this procedure can improve performance on the target domain:
> > > >
> > > > |                 | In-dist (real) | Out-of-dist (sketch) |
> > > > |-----------------|----------------|----------------------|
> > > > | ERM             | 82.5 (0.3)     | 35.9 (0.3)           |
> > > > | ...             | ...            | ...                  |
> > > > | SwAV            | 79.0 (0.3)     | 38.2 (0.4)           |
> > > >
> > > > We use the same procedure for self-supervised training using masked language modeling as well: we first initialize a pre-trained BERT model; then do continued pre-training using the unlabeled (target) data in the U-WILDS dataset; and finally finetune it on the labeled (source) data.
> > > >
> > > > We apologize for not having described this procedure clearly in the text. We have uploaded a new version of our revision that includes this in Section 5 (Algorithms) and Appendix B.4 (Self-supervision methods). Could we check if that answers your questions? We appreciate these comments and questions; thank you for helping us clarify the paper.
> > > >
> > > > ---
> > > >
> > > > > nevertheless, it is indeed surprising to know that semi-supervised and self-supervised models are quite good OOD generalizers. In that light, I would like to change my statements and agree that the experiment results are indeed significant. you should definitely add this discussion to the paper.
> > > >
> > > > Thank you! In our revision, as we’ve noted above, we’ve added the DomainNet results (which help to support this point); clarified the semi-supervised and self-supervised methods and why we’re evaluating them; and emphasized this discussion in the text. We appreciate the constructive suggestions and feedback.

---

> > > > > ### Comment · Reviewer_RpU9 · 2021-11-23
> > > > > **Response to authors**
> > > > >
> > > > > All your responses answer my questions! Thanks!

---

> > > > > > ### Author Response · Authors · 2021-11-24
> > > > > > **Thank you**
> > > > > >
> > > > > > Great, thank you for engaging with us on this and for helping to improve our paper!

---

### Official Review · Reviewer_3NYa · 2021-11-02

**Correctness:** 4
**Technical Novelty And Significance:** 2
**Empirical Novelty And Significance:** 3
**Recommendation:** 6
**Confidence:** 4

**Main Review:**

Strengths:
1) The problem of unsupervised domain adaptation is well-grounded in a practical learning setting where labels are scarce, but observations are not.  This makes the extension of the WILDS data set to this setting a very useful contribution that can facilitate impactful future work.
2) The empirical evaluation was fairly extensive in it's inclusion of a variety of unsupervised domain adaptation techniques.  I think it is important to validate the trend established in prior work that these techniques typically perform poorly in more realistic domain adaptation problems.  I also think the finding that data augmentation techniques work relatively well in image domains to be an interesting one that can motivate future work.
3) Perhaps a necessary consequence of this kind of work is that the paper serves as a nice survey of modern unsupervised domain adaptation, both in terms of data sets and methods.  Someone interested in the topic could use this paper to begin a literature search.

Weaknesses:
1) Ultimately, the novelty of this work is low.  The main novel contributions are the extension to the existing WILDS data set (using already established data sets used by the original WILDS authors) and some new empirical results using previously published results.  However, I do not think for this kind of work that novelty is paramount.
2) The descriptions of the data sets are relatively shallow.  Besides raw increase in instances, not much else is reported about the data sets.  Is there anything interesting to say about the instances that were chosen to be labeled by the original WILDS authors versus the ones added by U-WILDS?  Namely, is there anything unique about the new instances?  Can you characterize and measure how different the new instances are from the original ones?
3) Data preparation was not discussed at all.  The major practical concern I have about this endeavor is standardization across new and old instances.  If the original WILDS data set was released with some effort to process the data before repackaging and release, the U-WILDS data set would need to undertake the same process.  I would like to see some comments on whether effort was taken to ensure that the U-WILDS instances were preprocessed for consumption in the same way the WILDS data set were, or why that was not a necessary step.

**Summary Of The Paper:**

The authors introduce an extension to the new, but popular, data shift benchmark WILDS data sets called U-WILDS.  U-WILDS increases the number of instances in these data sets significantly (by a factor 3.5-14x times, depending on the data set), but includes no additional labels.  Such an extension allows unsupervised domain adaptation techniques to now be evaluated on the various WILDS data sets.  The authors evaluate a sampling of the state of the art in unsupervised domain adaptation on these new data sets.  The main finding is that most of the techniques performed poorly, except data augmentation in the image problems.  The authors suggest that this could motivate the need for better data augmentation techniques for other modalities.

**Summary Of The Review:**

While I recognize the low novelty of the work, I believe the contribution of the extended WILDS data set as well as the insights provided in the empirical evaluation are of significant enough value to the machine learning community to warrant acceptance.

---

> ### Author Response · Authors · 2021-11-18
> **Response to Reviewer 3NYa**
>
> We thank the reviewer for their feedback and suggestions. Below, we respond to the reviewer’s questions.
>
> ---
>
> > The descriptions of the data sets are relatively shallow. Besides raw increase in instances, not much else is reported about the data sets. Is there anything interesting to say about the instances that were chosen to be labeled by the original WILDS authors versus the ones added by U-WILDS? Namely, is there anything unique about the new instances? Can you characterize and measure how different the new instances are from the original ones?
>
> We agree with the reviewer on the importance of characterizing the unlabeled vs. labeled data, so that algorithm developers have a good understanding of the kinds of structure and leverage they might be able to use. In general, we strove to choose unlabeled data that was qualitatively similar to the existing labeled data: for example, in iWildCam, we obtained unlabeled data from camera traps that are drawn from essentially the same distribution as the labeled camera traps.
>
> We have attempted to provide these details in Appendix A, including describing the sources of the new unlabeled data; characterizing how it differs from the original labeled data in terms of domains; and describing the data processing. In our revision, we have edited Appendix A to add more detail on the differences between the unlabeled vs. labeled datasets for iWildCam (Appendix A.1), FMoW (Appendix A.3), MolPCBA (Appendix A.6), and Amazon (Appendix A.8). We thank the reviewer for bringing this important point up.
>
> ---
>
> > Data preparation was not discussed at all. The major practical concern I have about this endeavor is standardization across new and old instances. If the original WILDS data set was released with some effort to process the data before repackaging and release, the U-WILDS data set would need to undertake the same process. I would like to see some comments on whether effort was taken to ensure that the U-WILDS instances were preprocessed for consumption in the same way the WILDS data set were, or why that was not a necessary step.
>
> We also agree with the reviewer that it is important to be consistent in data preparation and to have a standardized protocol. We discuss our data preparation protocols for each dataset in Appendix A. In all cases, we made sure that the U-WILDS instances were preprocessed in the exact same way as the original WILDS instances. The only exception was for the Camelyon17 dataset, where the original labeled dataset was processed to be class-balanced; because we do not know the classes for the unlabeled data, we instead uniformly sampled patches at random for the unlabeled data.
>
> ---
>
> If the reviewer has any additional questions or concerns about the unlabeled vs. labeled data, data processing, or standardization that might be preventing them from providing a stronger recommendation, please let us know and we will do our best to answer them in this discussion period. Thank you.

---

> > ### Comment · Reviewer_3NYa · 2021-11-23
> > **Response to Authors**
> >
> > This was very helpful.  Thank you.  I would encourage the authors to have the pointer to Appendix A in the main body of the paper provide a little more insight into what specifically they discuss about the data sets in the appendix.  Other than that, I feel my questions were addressed.

---

> > > ### Author Response · Authors · 2021-11-24
> > > **Thank you**
> > >
> > > Thank you! That's a good suggestion. We can't edit the rebuttal revision right now, but we will do that for the subsequent version of this paper.

---

### Official Review · Reviewer_Dxvt · 2021-11-03

**Correctness:** 4
**Technical Novelty And Significance:** 1
**Empirical Novelty And Significance:** 4
**Recommendation:** 8
**Confidence:** 4

**Details Of Ethics Concerns:**

It would be good to:
- Verify that all of the datasets used have licenses that allow for public release, and that all datasets have been adequately anonymized/deidentified.
- Flag any potential fairness concerns with using these datasets for the benchmarking and selection of developed algorithms, e.g. as has been found in ImageNet [1].

[1] https://arxiv.org/pdf/2010.15052.pdf

**Main Review:**

The problem is well-motivated, and the use of realistic problems to benchmark unsupervised adaptation methods presents a major gain over prior datasets which rely on different stylized images (e.g. PACS). The datasets used span a wild variety of modalities and domains, and are fairly robust with large sample sizes. The paper is easy to read and easy to follow, and the experimental evaluations are robust.

I would suggest the following improvements:

- In order to present an informal upper bound for the performance of algorithms on the OOD domain, the authors should consider adding in two additional "oracle" baselines which have access to labelled OOD-domain data: 1) training an ERM model only on the OOD domain, 2) training an ERM model on all available data.

- The authors should justify why the OOD domain was the one selected in each dataset. If possible, the authors could consider allowing for the ability to swap around the various training/validation/test domains, though it might be impractical if some domains do not have sufficient data.

- The authors should clarify the model selection procedure for each of the algorithms. For example, how does each method make use of the Validation (ID) labelled examples versus the Validation (OOD) unlabelled examples for model selection?

**Summary Of The Paper:**

The authors propose U-WILDS, which extends the WILDS benchmark (typically used for domain generalization or subpopulation shift) to the unsupervised domain adaptation scenario. They select eight datasets from WILDS, spanning a variety of data modalities, and add in additional unlabelled examples from a variety of data sources. They benchmark a comprehensive set of algorithms which make use of unlabelled data, and find that most methods do not significantly outperform ERM, except for some limited cases which the authors characterize in detail.

**Summary Of The Review:**

Though the paper does not propose any novel methodology, I believe that it is a solid step towards the use of real-world datasets for benchmarking unsupervised domain adaptation algorithms, and would be a valuable contribution to the conference.

---

> ### Author Response · Authors · 2021-11-18
> **Response to Reviewer Dxvt, part 1/2**
>
> We thank the reviewer for their feedback and suggestions. Below, we respond to each in turn.
>
> ---
>
> > In order to present an informal upper bound for the performance of algorithms on the OOD domain, the authors should consider adding in two additional "oracle" baselines which have access to labelled OOD-domain data: 1) training an ERM model only on the OOD domain, 2) training an ERM model on all available data.
>
> Thank you for the suggestion! For four of the U-WILDS datasets (Amazon, CivilComments, iWildCam, FMoW), we created the unlabeled splits by acquiring additional data and then hiding their labels, so we were able to run fully-labeled ERM experiments using their ground truth labels. Specifically, we trained ERM models on all available training data (the original labeled data, as well as the “unlabeled” data with ground truth labels revealed), using the exact same held-out test set and evaluation protocol.
>
> For FMoW and iWildCam, where the challenge is primarily in performing well on unseen domains (that are not in the labeled training set), the fully-labeled ERM models obtained significantly higher OOD test performance, as shown in this table:
>
> |                         | ERM (original) | ERM (fully-labeled) |
> |-------------------------|----------------|---------------------|
> | FMoW (worst-region acc) | 34.8%          | 58.7%               |
> | iWildCam (macro F1)     | 32.2%          | 44.0%               |
>
> These results suggest that leveraging the unlabeled data could be helpful for these datasets.
>
> On Amazon and CivilComments, where the challenge is primarily in performing uniformly well over subpopulations of the training distribution, the fully-labeled ERM baselines show a more modest improvement in test OOD performance:
>
> |                                  | ERM (original) | ERM (fully-labeled) |
> |----------------------------------|----------------|---------------------|
> | Amazon (10th percentile acc)     | 54.2%          | 56.4%               |
> | CivilComments (worst-group acc)  | 66.6%          | 69.6%               |
>
> These results are consistent with prior observations that ERM models can have poor subpopulation performance even on large labeled training sets [1], necessitating other training procedures besides ERM for dealing with subpopulation shifts. We note that these are not quite oracle experiments, in the sense that the unlabeled data comes from the overall population, whereas the test performance is evaluated only on a particular subpopulation (like the worst 10th percentile group in Amazon, or the lowest-performing demographic group in CivilComments).
>
> In our revision, we have added these fully-labeled ERM results to Section 6 and Table 2, and we have also added Appendix E, which describes their setup in more detail.
>
> We note that the original WILDS paper [2] also contains oracle results for each of the datasets. The main difference is that the original WILDS paper did not use the data introduced in U-WILDS, which allows us to run fully-labeled ERM experiments using ground truth labels on the new data in U-WILDS.
>
> [1] Sagawa, S., Koh, P. W., Hashimoto, T. B., & Liang, P. (2019). Distributionally robust neural networks for group shifts: On the importance of regularization for worst-case generalization. arXiv preprint arXiv:1911.08731.
>
> [2] Koh, P. W., Sagawa, S., Xie, S. M., Zhang, M., Balsubramani, A., Hu, W., ... & Liang, P. (2021). Wilds: A benchmark of in-the-wild distribution shifts. International Conference on Machine Learning (pp. 5637-5664). PMLR.
>
> ---
>
> > The authors should justify why the OOD domain was the one selected in each dataset. If possible, the authors could consider allowing for the ability to swap around the various training/validation/test domains, though it might be impractical if some domains do not have sufficient data.
>
> For consistency, we kept the exact same test split (and therefore, the choice of OOD domains) as the original WILDS benchmark. We have edited Sections 3 and 4 to clarify this, and we have also emphasized it in Section 6.1. As described in the WILDS paper [2], the different datasets have different rationales for the choice of splits. In some cases, there were natural splits (e.g., for FMoW, where the domain shift is partially over time, the appropriate split would be to train on data from earlier time periods and test on data from later time periods). In other cases, the assignment of domains was simply done at random. For the PovertyMap dataset, the evaluation uses five folds, with each fold having a different training/validation/test split; however, PovertyMap is a relatively small dataset, and the original WILDS paper reported that doing something similar for the other datasets would have been too computationally expensive.

---

> > ### Author Response · Authors · 2021-11-18
> > **Response to Reviewer Dxvt, part 2/2**
> >
> > > The authors should clarify the model selection procedure for each of the algorithms. For example, how does each method make use of the Validation (ID) labelled examples versus the Validation (OOD) unlabelled examples for model selection?
> >
> > We apologize for the lack of clarity. None of our experiments use the Validation (ID) labeled examples or Validation (OOD) unlabeled examples for model selection; instead, they use only the Validation (OOD) *labeled* examples for model selection. This model selection procedure (using the Validation (OOD) labeled examples) is the same as in the original WILDS benchmark, which allows for direct comparisons to the results therein. We note that all of the following datasets are drawn from different distributions:
> > - Training examples
> > - Validation (OOD) examples
> > - Target (OOD) examples
> >
> > For example, in iWildCam, where the examples are photos and the domains are camera traps, the training domains, Validation (OOD) domains, and Target (OOD) domains are all disjoint sets of camera traps. This means that hyperparameter tuning on the Validation (OOD) data does not leak information on the Target (OOD) distribution. We have clarified this in Section 6.1 and Appendix D.3.
> >
> > In Section 7, we discuss how the availability of Validation (OOD) unlabeled examples in U-WILDS expands the range of potential model selection procedures and is an interesting avenue for future work. Another potential option would be to do unsupervised hyperparameter tuning directly on the Target (OOD) unlabeled examples, such as in [3].
> >
> > [3] Saito, K., Kim, D., Teterwak, P., Sclaroff, S., Darrell, T., & Saenko, K. (2021). Tune it the Right Way: Unsupervised Validation of Domain Adaptation via Soft Neighborhood Density. In Proceedings of the IEEE/CVF International Conference on Computer Vision (pp. 9184-9193).
> >
> > ---
> >
> > > It would be good to: Verify that all of the datasets used have licenses that allow for public release, and that all datasets have been adequately anonymized/deidentified.
> >
> > We can confirm that all of the datasets (both labeled and unlabeled) used have licenses that allow for public release. All datasets have been anonymized/deidentified. We have also ensured that the satellite image datasets (FMoW and PovertyMap) appropriately protect privacy. We have edited our Ethics statement to clarify these. We thank the reviewer for checking.
> >
> > ---
> >
> > > Flag any potential fairness concerns with using these datasets for the benchmarking and selection of developed algorithms, e.g. as has been found in ImageNet [1].
> >
> > In our Ethics statement, we have also attempted to discuss potential fairness concerns, especially with the CivilComments dataset, which deals with biases against mentions of particular demographic groups. We thank the reviewer for the pointer to the relevant reference; we have edited our Ethics statement to expand on this, and have included the reference as well as others.

---

> > > ### Comment · Reviewer_Dxvt · 2021-11-23
> > > **Thank you for the clarifications.**
> > >
> > > I thank the authors for the detailed response. All of my concerns have been addressed.

---

> > > > ### Author Response · Authors · 2021-11-23
> > > > **Thank you**
> > > >
> > > > That's great to hear. Thank you for all of the suggestions; they have been very helpful for us to improve the paper.

---

### Author Response · Authors · 2021-11-18
**Overall response, part 1/2**

We thank all of our reviewers for their thoughtful reviews, and we appreciate all of their encouragement and constructive feedback. Overall, reviewers found that the U-WILDS benchmark would be a useful and well-motivated resource for the community, and that the experiments were comprehensive. While we did not introduce any new methods in this paper, U-WILDS significantly expands the range of modalities, applications, and shifts available for studying unsupervised adaptation, which leads to the central conclusion of our work--- success on prior unsupervised adaptation benchmarks need not transfer to success on other benchmarks that reflect different real-world conditions. We believe that this is an important point for the ML research community to know about.

However, parts of the paper were unclear, which led to some questions from the reviewers. In this revision, we have edited the paper throughout to clarify and expand on the points of confusion that they raised, including:

- **Section 4, Datasets.** We have emphasized that U-WILDS uses exactly the same labeled datasets as in WILDS, and provides additional unlabeled data on top of that.
- **Section 5, Algorithms.** We have provided further context behind the use of self-training and self-supervision for domain adaptation tasks.
- **Section 6, Experiments.** We have expanded the experimental setup in the main text to clarify model selection, the use of unlabeled data, and other details. We have also included discussions of the new experiments below.
- **Section 7, Discussion.** We have elaborated on methods that can use domain annotations, such as multi-source domain adaptation algorithms.
- **Ethics statement.** We have expanded the Ethics statement to verify that the datasets have the appropriate licenses and have been adequately anonymized, as well as to further discuss potential fairness concerns.
- **Appendix A, Additional dataset details.** We have expanded on the discussion of the labeled vs. unlabeled data for the iWildCam, MolPCBA, and Amazon datasets.
- **Appendix B, Algorithm details.** We have elaborated on our discussion of domain-invariant methods to add more citations and discuss multi-source/multi-target methods, as well as to detail how we use self-supervision.
- **Appendix D, Experimental details.** We have clarified our model selection procedure.

For convenience, we have highlighted all of the substantive changes in the revision in green. We also made smaller line edits elsewhere in the paper in order to tighten the prose and fit the new additions within the page limit, but for clarity, we have omitted those highlights. In the individual responses below, we respond to each reviewer’s questions in more detail.

---

> ### Author Response · Authors · 2021-11-18
> **Overall response, part 2/2**
>
> Furthermore, we have added two new sets of experimental results in response to reviewer suggestions:
>
> 1. In response to a question from Reviewer RpU9 concerning the effectiveness of self-training and self-supervised methods on prior benchmarks, we have added **DomainNet results** for all of the methods that we had tested on U-WILDS. DomainNet is a standard domain adaptation benchmark for object recognition, and this is a replication experiment to show that the changes we made to the methods (e.g., standardizing the set of augmentations used) did not adversely affect their performance. Consistent with prior work, all of the methods we tested indeed outperformed ERM on our DomainNet experiments. Despite their success on DomainNet, these methods do not fare as well on the shifts in U-WILDS, which shows that **the success of these methods need not transfer across benchmarks**. In our revision, we mention these results in Section 6, and describe the setup and results in more detail in Appendix E.
>
> 2. On Reviewer Dxvt’s suggestion, we have added results on **fully-labeled ERM models** to four datasets (Amazon, CivilComments, FMoW, iWildCam). These models use ground truth labels on the “unlabeled” data, and can therefore provide an informal “upper bound” on how well a standard pseudo-labeling approach might do on these datasets. As we had created the unlabeled splits for these four datasets by acquiring additional data and then hiding their labels, we could train these fully-labeled ERM models by revealing the hidden labels on the “unlabeled” data and then training a model jointly on both the labeled and “unlabeled” data with ground truth labels.
>
>     On FMoW and iWildCam, where the challenge is primarily in performing well on unseen domains (that are not in the labeled training set), the fully-labeled ERM models do substantially better than the other methods which do not observe those ground truth labels. It is promising that this “upper bound” estimate is high, though we use the term “upper bound” loosely---it might also be possible to surpass this “upper bound” (even without access to ground truth labels) with methods that do more than standard ERM.
>
>     On Amazon and CivilComments, where the challenge is primarily in performing uniformly well over subpopulations of the training distribution, the fully-labeled ERM baselines show a more modest improvement. This is consistent with prior work on other datasets showing that even with a large amount of labeled data, ERM models can still obtain poor subpopulation performance compared to specialized methods such as distributionally robust optimization (DRO) techniques. However, these techniques have yet to be successfully adapted to the Amazon and CivilComments settings, and it remains an open challenge to develop similar methods that might work well on these datasets.
>
>     In our revision, we discuss these new results in Section 6 and Table 2, and describe the experimental setup in more detail in Appendix F.
>
> We are grateful to all of our reviewers for all of these suggestions. Please do not hesitate to contact us for any further questions or clarifications. Thank you!

---

### Decision · Program_Chairs · 2022-01-20

**Decision:**

Accept (Oral)

**Comment:**

This paper presents U-WILDS, an extension of the multi-task, large-scale domain-shift dataset WILDS. The authors propose an extensive array of experiments evaluating the ability of a wide variety of algorithms to leverage the unlabelled data to address domain-shift problems. The vision behind sounds quite ambitious and convincing to me, namely, the proposed U-WILDS benchmark would be a useful and well-motivated resource for the ML community, and their experiments were very comprehensive. Although they did not introduce any new methods in this paper, U-WILDS significantly expands the range of modalities, applications, and shifts available for studying and benchmarking real-world unsupervised adaptation.

The clarity, vision and significance are clearly above the bar of ICLR. While the reviewers had some concerns on the novelty, the authors did a particularly good job in their rebuttal. Thus, all of us have agreed to strongly accept this paper for publication! Please include the additional rebuttal discussion in the next version.